Manuscript prepared for Nonlin. Processes Geophys.
with version 2015/09/17 7.94 Copernicus papers of the LaTeX class copernicus.cls.
Date: 10 September 2016

# Compound extremes in a changing climate - a Markov Chain approach

Katrin Sedlmeier[1,*], Sebastian Mieruch[1,**], Gerd Schädler[1], and
Christoph Kottmeier[1]

[1]Institute for Meteorology and Climate Research, Karlsruhe Institute of Technology, Karlsruhe,
Germany
[*]now at: Federal Office of Meteorology and Climatology MeteoSwiss, Zurich, Switzerland
[**]now at: Alfred-Wegener-Institut, Helmholtz-Zentrum für Polar- und Meeresforschung,
Bremerhaven, Germany

*Correspondence to:* Katrin Sedlmeier
(katrin.sedlmeier@meteoswiss.ch)

**Abstract.** Studies using climate models and observed trends indicate that extreme weather has changed and may continue to change in the future. The potential impact of extreme events such as heat waves or droughts does not only depend on their number of occurrences but also on "how these extremes occur", i.e. the interplay and succession of the events. These quantities are quite unex-

plored, for past changes as well as for future changes and call for sophisticated methods of analysis. To address this issue, we use Markov chains for the analysis of the dynamics and succession of multivariate or compound extreme events. We apply the method to observational data (1951-2010) and an ensemble of regional climate simulations for Central Europe (1971-2000, 2021-2050) for two types of compound extremes, heavy precipitation and cold in winter and hot and dry days in summer.

We identify three regions in Europe, which turned out to be likely susceptible to a future change in the succession of heavy precipitation and cold in winter, including a region in south western France, northern Germany and in Russia around Moscow. A change in the succession of hot and dry days in summer can be expected for regions in Spain and Bulgaria. The susceptibility to a dynamic change of hot and dry extremes in the Russian region will probably decrease.

## 1 Introduction

Multivariate extreme events (in this paper used in the sense of extremes of two or more climate variables occurring simultaneously) are likely to impact society greater than their univariate counterparts. For agriculture for example, the impact of a heat wave and a drought occurring at the same time is higher than for a univariate extreme where the other variable is in a normal state. These

multivariate or so called compound events (IPCC, 2012) have received more and more attention in the scientific literature over the past years although still not to the extent of extremes of only one variable. Methods to analyze them include simple threshold analysis, multivariate distribution func-

tions using copulas (e.g. Schoelzel et al., 2008; Durante and Salvadori, 2010), Bayesian approaches (e.g. Tebaldi and Sansó, 2009) or indices which are derived from multiple variables (e.g. the wild-fire index KBDI (e.g. Keetch et al., 1968) or the revised CEI Gallant et al. (2014)). Furthermore, methods of multivariate extreme models have been used for the geostatistical analysis of spatially distributed extremes (Neves, 2015). All these methods focus mostly on the linear climate change signal - the absolute change in the number of occurrences or the calculation of return periods. The succession, i.e. the temporal ordering of the compound events is in most cases not the main objective. For instance, the IPCC (IPCC, 2012) states : "*A changing climate leads to changes in the frequency, intensity, spatial extent, duration, and timing of extreme weather and climate events, and can result in unprecedented extreme weather and climate events.*" What is implicitly addressed with "duration and timing", but not explicitly stated is the succession of extreme events, which is quite unknown for past as well as future extremes.

The method proposed here, which is based on Markov chains, concentrates on the dynamical be-havior or succession of these compound extreme events and studies an aspect of climate change which has not received much attention up to now, but is nevertheless important. We investigate a behavior of extremes which cannot be determined by simply analyzing the changes in the number of extremes. We can, for example, reveal changes in the entropy of the succession of compound ex-tremes which is connected to the chaotic behavior of the climate variable. Thus an observed increase of this measure could be connected with an increase in the chaotic, intermittent or irregular nature of the system. On the other hand, a decrease of entropy corresponds to a slow-down of these dynamics. Knowledge about such developments for future climate, which rarely exists, could be important for many sectors e.g. agriculture, economy and society.

Previous studies on model dynamics have concentrated more on overall dynamical behavior such as Steinhaeuser and Tsonis (2014) who have conducted a model intercomparison study focusing on dynamical aspects based on a climate networks framework. The method introduced in this paper is inspired by the work of Mieruch et al. (2010). The idea is to understand climate time series as trajectories on a complex, possibly strange attractor (Lorenz, 1963). We partition the time series or state space into a finite number of states. This yields a coarse-grained description of the system, which can then be analyzed in the framework of symbolic dynamics (Ebeling et al., 1998; Daw et al., 2003). We apply a Markov Chain analysis on these symbolic sequences representing compound extremes, and characterize their dynamical or successional behavior using a small set of descriptors.

In this paper we study two different kinds of compound extreme events which are likely to have an impact on society, namely cold and heavy precipitation in winter, and heat and drought in summer. The Markov method is applied to E-OBS observational data (1951-2010) (Haylock et al., 2008), and an ensemble of regional climate simulations with the regional climate model COSMO-CLM driven by different global climate model data and ERA-40 reanalysis (Uppala et al., 2005). The time periods considered are the recent past (1971-2000) and the near future (2021-2050).

We identify regions in Europe, where the dynamical behavior of the analyzed compound extremes is prone to change. These findings highlight that it is not only the (simple)

linear increase of the occurrence of extremes (due to an increase in mean and variability), which is a challenge for adaption and mitigation. On top of these changes the regions also have to struggle with changes in the succession of compound extremes (defined as relative to a new normal state with

changed mean and variability).

     The strategy of this study is first to show that the Markov method is able to extract different dynamics of compound extremes for different regions in Europe, based on observational data and model data. Thus, on the one hand we see that the method yields meaningful information and on the other hand we show that the climate models are able to reproduce these dynamics in the frame

of acceptable uncertainties. Additionally, we extract temporal change signals of the dynamics of compound extremes based on observations between the periods 1951-1980 and 1981-2010. This information is new and if used as supplementary information to other analyses, could lead to a better understanding of changes of extremes in Europe. For this paper, the magnitude of the observed past changes have been assessed, because it is important for a better interpretation and classification of

future changes which are calculated by using the simulated regional climate model data. A comparison of the change signals between 1971-2000 and 2021-2050 to the observed past changes shows that they are of the same order of magnitude.

     The paper is divided into the following sections. In Sect. 2, data and method will be introduced, followed by a sensitivity analysis of the method with respect to spatial and temporal variability as

well as the error of estimation using FT surrogates in Sect. 3. A validation of the model ensemble is shown in Sect. 4. The change signal is analyzed in Sect. 5. Summary and outlook will be given in Sect. 6 and some areas discussed where the application of this method might be of value.

## 2   Data and Methods

### 2.1   Regional Climate Ensemble

For our analysis, we use a 12-member ensemble of regional climate simulations for Central Europe at a resolution of 50 km. The ensemble has been generated by downscaling different global climate model outputs with the regional climate model COSMO-CLM (COnsortium for Small scale MOdelling model - in CLimate Mode, Doms and Schättler (2002); Rockel et al. (2008)), further referred to as CCLM. The CCLM is a non-hydrostatic climate model coupled to the soil vegetation

model TERRA and is the climate version of the numerical weather model of the German weather service. Data from six different global climate models (GCMs) has been used as initial and boundary data. Two of the GCMs have used the emission scenario A1B (Nakicenovic and Swart, 2000) as external forcing : CCCma3 (Scinocca et al., 2008) and three realizations of ECHAM5 (Roeckner et al., 2003). The other four, ECHAM6 (Stevens et al., 2013), CNRM-CM5 (Voldoire et al., 2013),

HadGM3 (Collins et al., 2011) and EC-EARTH (Hazeleger et al., 2010) have used the emission scenario RCP8.5 (Riahi et al., 2011; Van Vuuren et al., 2011). Additionally the Atmospheric Forcing Shifting method (Sasse and Schädler, 2014) was applied to the ECHAM6 data. For this method the global climate data interpolated to the 50 km grid is shifted by two grid points in all cardinal directions before being used as boundary data. This accounts for the uncertainty in positioning of
synoptic systems when interpolating the GCM data to the required resolution for forcing the RCM simulations. As all five ECHAM6 driven simulations obtained this way exhibit a high correlation, they are all weighed with a factor of 1/5 when calculating the mean. All other models receive a factor of one which leads to an effective ensemble size of eight. Additionally we use a COSMO-CLM run driven by ERA-40 (Uppala et al., 2005) boundary conditions. The ERA-40 reanalysis boundary
conditions are assumed to be close to the "'true"' observed state. Nevertheless, they depend on the model, observational data and assimilation tequnique, among others and are not free from biases (see e.g. Hagemann et al., 2005; Simmons et al., 2004).

The simulation time periods are the recent past (1971-2000) and the near future (2021-2050). An
analysis of the temperature trends of different ensemble members showed that the distribution of trend depends more strongly on the chosen global climate model than on the emission scenario. We therefore combine simulations with boundary data from GCMs with different emission scenarios to set up our ensemble.

We choose six regions, each comprising $6 \times 6$ grid points for our analysis. The regions are chosen
based on the PRUDENCE regions (Christensen and Christensen, 2007) which could not be used because of the necessity of the same amount of grid points for each area, and due to test results which show a different behavior for these regions. We investigate 30 year periods of daily data, thus each time series consists of $\approx 11,000$ data points, yielding $\approx 36 \times 11,000 \approx 400,000$ points in time for each region and ensemble member. The model domain and the six investigation areas, which are
located in Spain, France, Germany, Scandinavia, Bulgaria, and Russia are shown in Fig. 3. These roughly match the PRUDENCE regions which are not applicable for the analysis since equal sized areas are a requirement for comparison among regions.

## 2.2 Observational data

For the comparison of our regional climate ensemble with observations, we use temperature and
precipitation data from the gridded E-OBS dataset (Haylock et al., 2008). This dataset was produced as part of the ENSEMBLES project by interpolating station data from the ECA&D station dataset (European Climate Assessment, Klok and Klein Tank, 2009) to a 25 km grid. The station density is highest in Switzerland, the Netherlands and Ireland and rather low in Spain and the Balkans which leads to an over-smoothing in these areas. This especially affects extremes and has to be taken into
account when validating our ensemble against E-OBS data. Furthermore it should be noted that a

**Table 1.** ECA&D Station data

|   | Station (Station number) |
|---|---|
| 1 | Bamberg (40) |
| 2 | Hamburg Fuehlsbuette (47) |
| 3 | Hohenpeissenberg (48) |
| 4 | Potsdam (54) |
| 5 | Hamburg-Botanischer-Garten (4180) |
| 6 | Hamburg Sankt-Pauli (4184) |
| 7 | Hamburg-Wandsbek (4186) |
| 8 | Quickborn Kurzer Kamp (4536) |

comparison of E-OBS and another gridded dataset, namely Hyras (Rauthe et al., 2013) (only Central Europe), with respect to the dynamical behavior that we analyze in this paper, revealed differences between the two datsets (Sedlmeier, 2015). A comparison of dynamical aspects of different observational datasets yields an interesting application of the method which however will not be addressed within this paper. We additionally use blended temperature and precipitation time series starting from 1900 of eight stations (all in Germany) of the ECA&D dataset for a sensitivity analysis described in Sect. 3. The eight stations are listed in Tab. 1.

### 2.3 Compound extremes with Markov Chain descriptors

The method used in this paper consists of describing temperature and precipitation time series by a Markov Chain and subsequently calculating descriptors, which characterize the dynamical (successional) behavior of the compound extreme states. The method has been used in biology (Hill et al., 2004) to describe dynamics of succession of species in a rocky subtidal community. It has been introduced to atmospheric science by Mieruch et al. (2010) who used it for climate classification and a comparative study of two regions. In this section, a short introduction to Markov chains is given, followed by a step by step description of the method.

A first order, $m$ state ($m$= number of discrete states of the Markov Chain), homogeneous Markov Chain is a time discrete, state discrete stochastic process which fulfills the Markov property:

$$P(x_t|x_{t-1},x_{t-2},...,x_{t-n}) = P(x_t|x_{t-1}) \tag{1}$$

meaning that the present state $x_t$ is only dependent on the preceding state $x_{t-1}$. From the Markov chain, a transition probability matrix $\mathbf{P}$ of the order $m \times m$ can be calculated which consists of all possible conditional probabilities $P(x_t|x_{t-1})$ between the $m$ different states of the Markov chain.

For a homogeneous ($\equiv$ stationary) Markov chain, the transition probability matrix is time independent. A stationary distribution $\boldsymbol{\pi}$ is a vector that fulfills the following equation

$$\boldsymbol{\pi} = \mathbf{P}\boldsymbol{\pi}. \tag{2}$$

To test for homogeneity one must solve the eigenvalue problem of equation 2 to calculate the stationary distribution $\boldsymbol{\pi}$. If this is identical to the empirical distribution

$$\hat{\pi}_j = \frac{n_j}{\sum_j n_j}. \tag{3}$$

the time series is considered stationary. The entries (transition probabilities) of the transition matrix $\mathbf{P}$ are estimated by

$$\hat{p}_{ij} = \frac{n_{ij}}{\sum_i n_{ij}}. \tag{4}$$

In the following, the main steps of the Markov analysis are explained:

a) **Partitioning and combining of univariate time series to a multivariate symbolic sequence**
To represent the univariate time series (here daily mean temperature anomalies and daily precipitation anomalies) by a Markov chain, each time series is partitioned into a symbolic sequence of extreme and non-extreme regimes. These univariate symbolic sequences are then combined into a multivariate symbolic sequence of $m = 2^v$ different states ($v$ number of variables). In this paper, $v = 2$, thus there are four possible states.

b) **Calculation of the transition probability matrix**
From the $2^v$-state Markov chain, a transition probability matrix $\mathbf{P}$ of dimension $2^v \times 2^v$ can be calculated. Two conditions have to be met when calculating the descriptors. No entry of the transition probability matrix should be equal to zero and the time series needs to be stationary for the transition probability matrix to be time independent (see equations 2, 3).

c) **Calculation of the descriptors**
Following Mieruch et al. (2010), we focus on only three of the descriptors mentioned in Hill et al. (2004): persistence, recurrence time and entropy. These descriptors can be estimated for single states of the symbolic sequence or for the whole system. As the focus of this work lies on the compound extreme state, only the single-state definition of the descriptors is considered.

**Persistence:**

$$P_j = \hat{p}_{jj} \tag{5}$$

The persistence gives the probability that the system will stay in an extreme state in the following time step if it resides in an extreme state at the current time step. The limits are 0 (the

system will never stay in the extreme state) and 1 (the system will always stay in the extreme state). Regarding the succession of the compound extremes, the persistence tells us how long the extremes last.

**Recurrence time:**

$$R_j = \frac{1 - \hat{\pi}_j}{(1 - \hat{p}_{jj})\,\hat{\pi}_j} \tag{6}$$

The recurrence time describes the number of days the system needs to get back to the extreme state. The limits are 0 (the system never leaves the state, corresponding to a persistence of 1) and $\infty$ (the system never comes back to the extreme state). The recurrence time is connected to the persistence. If the persistence increases, the recurrence time will also increase and vice versa, except if a change in the number of states $\hat{\pi}_j$ occurs. Thus, it is important to include the absolute number of the states for the interpretation of the results.

**Entropy:**

$$H(p_j) = -\sum_i \hat{p}_{ij} \log \hat{p}_{ij} / log\left(\frac{1}{m}\right) \tag{7}$$

According to Shannon (1948), the entropy is an inverse measure of the predictability of the Markov Chain. Its limits are 0 (deterministic system) and 1 (random system). The dynamics of complex chaotic systems lie in between these limits, thus the entropy can give a hint to underlying complex dynamics like deterministic chaos, which is not possible with standard linear methods. To really test for deterministic chaos other methods, based on state space reconstruction (e.g. estimating the correlation dimension, Lyapunov exponents etc.) to find strange attractors, are more suitable. Thus, in the sense of successive compound extremes a change in entropy tells us if the succession of extreme states gets more chaotic or more regular.

d) **Data pre-processing**

In order to extract the information on successive compound extremes, we have to remove linearities (e.g. trends) and cycles, which would bias the results. Thus, we remove the external solar forcing by subtracting the mean annual cycle. A long-term trend is removed by a linear regression. Although, e.g. the temperature trend due to the anthropogenic $CO_2$ emissions is removed from the data, we hypothesize that all changes in the succession of extremes are linked to the $CO_2$ increase. The reason for this is that the $CO_2$ forcing is the only difference between the model runs for the periods 1971-2001 and 2021-2050.

We use percentiles to partition our datasets, and keep the number of univariate extreme events the same for different time periods and regions as well as for all ensemble members. By this,

the results can be compared among each other, differences are only due to different dynamical behavior. For partitioning dry days, we did not use precipitation anomalies but the effective drought index (EDI). The EDI (see Sect. 2.4) is related to soil moisture and is therefore a much better measure for describing dry extremes than precipitation itself, since all percentiles below
the percentage of dry days will lead to the same partitions.

In order to get a better feeling for the descriptors and understand how they relate with each other, we will do a small thought experiment. We take a Markov chain consisting of a time series of 1000 symbols of which 10% are extreme, the rest are normal. In this case a persistence of 0.5 would mean that in half of the 100 extreme cases, the next case is also extreme, there are 50 transitions
from the extreme state to the extreme state. The maximum episode length in this case is thus 51 extreme states in a row (with all others randomly distributed). The recurrence time and entropy are inversely related to how these 50 extreme transitions are ordered. Recurrence time depends on the number of episodes (fewer episodes lead to a larger recurrence time, more episodes to a shorter recurrence time) and entropy additionally on the mean episode length. In this paper, we also look at
changes in the descriptors. A change in persistence of 0.05 in the above case would mean 5 more extreme-extreme transitions per 1000 days, and an increase from 50/100 to 55/100 (extreme-extreme transitions/extreme-normal transitions) is surely a noticeable change. The range of actually probable values of the descriptors is smaller than the whole possible range. A persistence of 0.99 for example, would mean that there is only one extreme episode in the whole time period, all 100 extreme states
occur after each other. In a climate system, this is unlikely to happen. Thus, for climate one cannot expect to observe a change of the daily persistence from e.g. 0.5 to 0.8, because such a change would be catastrophic.

### 2.4    Effective drought index: EDI

The effective drought index (EDI) is an index for detecting drought conditions by calculating daily
deviations of precipitation from a climatological mean state. It was proposed by Byun and Wilhite (1999). An important concept of the EDI is the use of effective precipitation EP, rather than precipitation P itself. EP describes the depletion of water sources by a weighted summation over the 365 days preceding a given day d:

$$EP_d = \sum_{n=1}^{365} \left( \frac{\sum_{m=1}^{n} P_{d-m}}{n} \right) \tag{8}$$

By this, the memory effect of the soil is taken into account. EP therefore strongly correlates with soil moisture and the EDI is thus a good measure when considering droughts. Using the effective precipitation EP, the EDI is calculated by the following formula for a given day d:

$$EDI_d = \frac{EP_d - \overline{EP_{d,rm}}}{\sigma \left( EP - \overline{EP} \right)_d} \tag{9}$$

where $\overline{EP_{d,rm}}$ is the climatological mean corresponding to a given day d calculated as the 30-year average over a 5 day running mean (rm=5). By subtracting this climatological mean of EP from the daily value, the yearly cycle is removed from the EDI time series.

### 2.5 The Markov descriptors for two compound extremes

To calculated the Markov descriptors we first calculated temperature and precipitation anomalies using the mean annual cycle of the respective time period and ensemble member/observation. We calculate the Markovian descriptors for for two types of extremes

- cold and heavy precipitation (temperature anomaly ($T$) < 10th percentile and precipitation anomaly ($P$) > 75th percentile) in winter (DJF) and

- heat and drought ($P$ >90th percentile and EDI < 25th percentile) in summer (JJA),

and for the six regions shown in Fig. 3. As an example, we show how we constructed the Markov chain for the cold/heavy precipitation extreme at a single grid point. First we identify temperature values below the 10th percentile $T_{\ell,t}$ and above $T_{h,t}$ ($t$ is the time index). Similarly we identify low and high precipitation values $P_{\ell,t}$ and $P_{h,t}$. Following we combine these symbols and find the following possible states: ($T_{\ell,t}$,$P_{\ell,t}$), ($T_{\ell,t}$,$P_{h,t}$), ($T_{h,t}$,$P_{\ell,t}$) and ($T_{h,t}$,$P_{h,t}$). Now we can rename these states to e.g. $S_{\ell,\ell,t}$, $S_{\ell,h,t}$, $S_{h,\ell,t}$ and $S_{h,h,t}$ and then a Markov chain could look like: $S_{\ell,h,1}, S_{\ell,\ell,2}, S_{\ell,\ell,3}, S_{h,h,4}, S_{h,h,5}, S_{h,h,6}, S_{h,\ell,7}, \ldots, S_{h,h,N}$, where $N$ is the total number of data points. From such a sequence we calculate the transition probability matrix and from this, the descriptors.

## 3 Sensitivity analysis

Before applying the method to the observational data and the model ensemble, we tested the applicability of the method by several sensitivity tests using the above defined descriptors. Therefore we consider the gridded E-OBS data and additionally ECA&D station data.

### 3.1 Spatial variability

In order to test the spatial variability of the descriptors we calculated them for the entire E-OBS dataset for the time period 1971-2000 for the two types of extremes mentioned above.

The descriptors were calculated for each grid point but taking into account not only this grid point but additionally the eight neighbouring gridpoints thus using a moving window if 9 gridpoints. The time series for each grid point were detrended and partitioned separately before the 9 partitioned time series were merged to calculate the descriptors. The reason for using this moving window of 9 gridpoints is the fulfillment of the criteria of the Markov method (see Sec.2.3) which is not given

for the entire area when using only single gridpoints. Using this moving window does not alter the general spatial pattern and smoothness of the results.

For both type of extremes, the descriptors show smooth spatial patterns (see Fig. 1), nevertheless variations between different regions can be identified.

The persistence for the winter extremes (left side of Fig. 1) is lower than for summer extremes,
especially in northern and Central Europe compound cold and wet events are most likely events of a short duration and rather rare (with recurrence times of up to 400 days). Along the Mediterranean coast and south eastern Europe the values are higher and probabilities of residing in a compound extreme state of over 50% are observed. The recurrence time for these events is also comparatively low (around 100 days). Interpreting the results, one has to keep in mind that we are always referring
to relative compound extremes. The entropy is around 0.9 for most of the area with small regions showing lower entropies down to 0.5. These high values can be explained by the low persistence - as compound winter extremes are grouped in very short episodes (low persistence), they are very hard to predict. The highest persistences for summer events (right side of Fig. 1) are observed in Scandinavia and the eastern part of the E-OBS domain and lowest in Central Europe and the northern coast of
Spain. The persistence is above 50% for the whole domain, which means that the probability of the system residing in a compound extreme state is high and these events are grouped in episodes of long duration. The recurrence time lies between 40 and 100 days and is as such also lower than that for compound winter events. Lowest values are observed in the Balkan region. The entropy lies between 0.4 and 0.65 which means that the extreme events are not so easy to predict, especially for
parts of Central Europe where the entropy is highest. However, according to our definitions, summer extremes can better be predicted than winter extremes.

For the main analysis in this paper we apply the method to 6 regions which we chose in rough agreement with the Prudence regions. The crucial point for being able to compare the descriptors of different regions is that each region contains the same amount of grid/data points. Since the
descriptors do not vary strongly within the Prudence regions, we chose regions consisting of 6x6 gridpoints from within these widely used regions. The regions which will be analyzed in the further sections of this paper are shown in Fig. 3.

Note that the results shown in Fig. 1 can only be qualitatively compared to those of the regions considered later or the station data in the next section as the number of grid points (or stations)
contributing to the analysis differs.

### 3.2 Temporal variability

To assess the temporal variability of the descriptors we calculated the descriptors for 30-year moving windows of observational station data from the ECA&D station dataset Klein Tank et al. (2002). Since we are interested in the daily values of temperature and precipitation, only stations were chosen
with a continuous daily record (with an allowance of 50 missing values at most). Using these criteria

there are eight stations with temperature and precipitation time series from 1900-2015 of which all are in Germany (see Tab. 1). One station has 15 missing values for temperature. These days were excluded from the analysis, considering the 30year time windows consisting of 10950 days, this amounts to roughly 0.1% of the values and does not alter the value of the descriptors. Of these eight stations, 5 are in the vicinity of Hamburg and have the same values for the first 17-22 years. The records of two stations in Hamburg are identical throughout the whole time period, therefore only one of them is included in the analysis which leaves a total of seven stations.

The descriptors were again calculated for both types of compound events. Linear temperature trends were removed separately for each of the 30year time-windows and in order to fulfill the criteria of the Markov method (stationarity and non-zero entries of the transition probability matrix, see Sec. 2.3), the partitioned data of these seven stations were combined to one time series to calculate the descriptors.

The results are shown in Fig. 2 for both winter (black) and summer (gray) extremes. Especially for the persistence and recurrence time, a clear shift is visible between 1930 and 1950. This time range is not preindustrial, but the crucial point is that the observed shift coincides with an observed shift in the global increase in $CO_2$ around 1950 (see e.g. Pachauri et al., 2014, Fig. SPM.1 (d)). From this finding we observe two main points:

1. The descriptors (especially persistence and recurrence time) seem to be sensitive to changes of the $CO_2$ increase. That means a stronger increase of $CO_2$ (e.g. from 1950 on) yields a lower level of persistence and to a higher level of recurrence time. Although $CO_2$ is still increasing after 1950, the recurrence time e.g. remains constant. Hence, the recurrence time seems not to be dependent on the absolute $CO_2$ concentration, but on the increase of latter.

2. Thus we can conclude that the natural variability of the descriptors can be approximated by the variability observed before and after the shift. This natural variability is smaller than the shift of the mean.

Concluding, due to the non-availability of preindustrial data we could not really test the natural variability of the descriptors in preindustrial times. But we could show that the approximate natural variability (before and after the shift in 1950) is smaller than the shift, which is probably due to the change in $CO_2$ increase. Just for a rough estimation: the mean level shift of the persistence for winter extremes is about 50% (from 0.2 to 0.1) and for the recurrence time it is about 20% (from 180 to 140 days). Regarding our results of changes of the descriptors (1971-2000 vs. 2021-2050) presented below (see see Sect. 6), we find changes of the persistence larger than 50% and changes of the recurrence time larger than 20%. We additionally perform significance tests on our results which show that these changes are indeed significant, excluding natural variability as the source for the observed changes.

### 3.3 Error of estimation using Fourier Transform (FT) surrogates

To assess the estimation error of the descriptors we used the Multivariate Iterated Amplitude Adjusted Fourier Transform (MIAAFT) algorithm as described by Venema et al. (2006); Schreiber and Schmitz (2000). By this algorithm, the data is shuffled and thus the original distribution is preserved. In addition, the auto and cross-correlation of the temperature and precipitation time series are approximately preserved. We constructed 100 MIAAFT surrogates for the temperature and precipitation anomalies (or the EDI time series for summer events, respectively) for the E-OBS dataset for the reference period (1971-2000). We then estimated the standard deviation of the descriptors calculated from these surrogate time-series. It is important to note that this standard deviation, under the framework of such a bootstrap test, already represents the standard error of the mean, which corresponds to the normal standard deviation devided by $\sqrt{N}$. The errors for both types of extremes and the six regions are listed in Tab. 2. The errors do not vary much between the different extremes and regions, the error of the persistence is in the order of 0.01 or lower, the one of the recurrence time between 1 and 2.6 and the error of the entropy in the order of 0.005. Adopting these errors to the values of the E-OBS descriptors for the reference period (shown in Figs. 4 and 5 in Sect. 4) the error of the persistence is about 2-10%, for the recurrence time about 2% and for the entropy about 1-2% (cf. Tab. 2).

This estimation error is much smaller than the ensemble uncertainty and can approximately be neglected. This shows that the estimation of the descriptors is robust. Further, we will consider the E-OBS data approximately as truth and we will use the ensemble uncertainty as the error for our main analysis.

**Table 2.** Estimation of the error of the descriptors by using MIAAFT surrogates for winter and summer extremes. The values were calculated using the E-OBS dataset for the reference period (1971-2000). In parentheses the percentage of the error with respect to the value of the E-OBS descriptors for the same time period and region are given.

|      | DJF | | | JJA | | |
|------|------|------|------|------|------|------|
|      | P | R | E | P | R | E |
| reg1 | 0.010 (7.9%) | 1.701 (2.1%) | 0.004 (0.5%) | 0.007 (1.2%) | 1.183 (1.8%) | 0.009 (1.6%) |
| reg2 | 0.011 (8.2%) | 2.182 (1.5%) | 0.010 (1.0%) | 0.010 (1.7%) | 2.055 (3.5%) | 0.010 (1.7%) |
| reg3 | 0.010 (7.9%) | 2.563 (1.5%) | 0.005 (0.6%) | 0.009 (1.6%) | 0.923 (1.4%) | 0.007 (1.1%) |
| reg4 | 0.008 (12.3%) | 1.150 (1.0%) | 0.005 (0.6%) | 0.008 (1.3%) | 0.990 (1.8%) | 0.011 (1.9%) |
| reg5 | 0.010 (3.9%) | 2.450 (1.0%) | 0.010 (2.2%) | 0.008 (1.3%) | 1.103 (1.7%) | 0.009 (1.5%) |
| reg6 | 0.007 (1.8%) | 0.797 (1.4%) | 0.004 (0.4%) | 0.009 (1.6%) | 1.150 (2.0%) | 0.009 (1.7%) |

## 4 Markovian descriptors for the reference period 1971-2000

Fig. 4 shows the descriptors for cold extremes and heavy precipitation in winter from 1971-2000. As for all boxplots in this chapter, the boxes show the 25th and 75th quantile of the ensemble (interquartile range) and the whiskers the minimum and maximum value of the ensemble. The colored line marks the ensemble median and the gray line the ensemble mean. Crosses mark the descriptors of the observations. The observed persistence for the different regions lies between 0.06 and 0.37. This means that the system does not stay in this extreme state for a very long time, the lowest observed persistence is in region 4 (Scandinavia) where extreme-extreme transitions are very rare. The recurrence times vary strongly between the regions, the values are between 64 and 314 days. Regions 1 and 6 (Spain and Bulgaria) show the lowest recurrence times. In region 6 (Bulgaria) the compound cold and wet episodes have the longest duration and occur with the highest frequency. The entropy of the observations lies between 0.86 in region 3 (Germany) and 0.96 in region 1 (Spain) and between 0.74 in region 3 (Germany) and 0.98 in region 1 (Spain) for the CCLM ensemble. Thus, the deduced entropy (both, observations and model) covers a rather small portion of the range of theoretically possible values from 0 to 1. As mentioned in Sect. 2.3 the range in which we actually expect the values of the descriptors is smaller. Therefore, when comparing the descriptors, the values have to be interpreted relative to the regions. One must be careful, however, because the descriptors do not permit to draw any conclusions about the absolute predictability of the states as long as the total numbers of states are not considered.

Focusing on the descriptors for the CCLM ensemble (box plots and gray bars in Fig. 4), we can see that with this method we are able to detect significant differences in dynamical behavior between some of the regions. In comparison to the descriptors of the observations (crosses in Fig. 4), the ensemble is able to capture the differences between the regions fairly well except for the persistence in region 5 where the ensemble shows a much lower persistence and the recurrence time of region 4 (Scandinavia) which is lower for the observations. However, these are regions where the density of station data underlying the E-OBS dataset is not very high and the E-OBS results may not be as reliable. The highest persistence is again in region 6 (Bulgaria) which also shows the lowest recurrence time and therefore has comparatively long events which occur more frequently than in other areas. The triangles mark the descriptors of the reanalysis driven simulations. They fit well for some regions, for others they are farther away from the observations than the CCLM-ensemble.

Fig. 5 shows the descriptors for hot and dry extremes in summer. Crosses again mark the descriptors of the observations. Persistence and recurrence time are higher, entropy is lower for hot and dry summer extremes than for cold and wet extremes in winter. A direct comparison can be maid because the extreme were partitioned such that the number of univariate extremes is the same for hot and dry extremes and cold and wet extremes. This might partly be due to the lower variability of EDI compared to precipitation anomalies but one would also expect the dynamical behavior of these extremes to be different. By our definition, hot and dry episodes in summer are longer and not as

frequent as cold and wet extremes in winter. The highest persistence is in regions 4 and 5 (Scandinavia and Russia), the lowest in region 3 (Germany). The entropy lies between 0.53 and 0.60 and is highest in region 2 (France) and lowest in region 6 (Bulgaria). The values are lower than for the cold and wet extremes, the winter compound extreme state exhibits more complex dynamics and is harder to predict (caution: this is also influenced by the total number of extremes). The CCLM ensemble (box plots) again captures the tendencies of the observed descriptors fairly well but shows a large spread and differences between the regions are mostly not significant for persistence and recurrence time. The ERA-40 driven CCLM simulations (triangles in Fig. 5) again fit well to the observations for some regions and show very different behavior for others.

For both types of compound extremes the ensemble mean and median seem to be able to capture the differences between regions shown by observations although not always in absolute numbers. An interesting result is that reanalysis driven CCLM data is sometimes farther away from the observational descriptors than the model data, especially for the cold and wet extremes in winter. This leads to the question whether the dynamical behavior of the driving GCM is greatly altered by the RCM downscaling and errors in both models compensate during the downscaling process. A further cause of this deviation of the ERA-40 driven simulations could be a misrepresentation of the dynamics by the reanalysis dataset. A follow up study comparing dynamical behavior of both RCM and GCMs is planned for the future. Additionally it would be interesting to also compare different reanalysis datasets using this method as there have been studies showing differences in their variability [e.g.][]hagemann2005.

## 5 Climate change signal of the Markovian descriptors

### 5.1 Change signal within the reference period

In order to get an idea about the order of magnitude of the change signal, the observational E-OBS dataset was split into two equal parts of 30 years, 1951-1980 and 1981-2010. The descriptors were calculated for both time periods and a change signal derived.

For cold and wet extremes (see Fig. 6) all regions except France show a decrease in persistence, region 5 and 6 (Russia and Balkan) show the strongest absolute decrease ($\approx 0.15$) and Germany the highest relative decrease of -72 % (relative changes are shown above the respective bars). The recurrence time does not change much for all regions except region 5 (Russia) where it decreases by 150 days. In this region, compound cold and wet extremes occurred more frequently but were of shorter duration in 1981-2010. The entropy shows a decrease of more than 5% in Spain and Germany where the system becomes more regular. In Spain an increase of entropy is observed and the compound extremes are harder to predict in 1981-2010 with respect to 1951-1980. The change signal for all descriptors and seasons (except for the entropy of France and Russia) are greater than the estimated error by FT-surrogates (see Tab. 2), thus these changes are robust.

Changes for hot and dry extremes in summer (see Fig. 7) are below 10% for most regions. Nevertheless for most regions these changes are still greater than the estimated errors by FT-surrogates (see Tab. 2). In Scandinavia, both persistence and recurrence time show a decrease, the extreme episodes are of shorter duration but occur more often. In Spain and Germany, both descriptors show an increase - especially in recurrence time, thus episodes of compound extremes occur less frequently. An increase in recurrence time can also be seen in Russia. The entropy increases in regions 3-6 (Germany, Scandinavia, Russia and Balkan), in these regions the system becomes less regular with respect to compound hot and dry events and harder to predict, whereas in Spain the Entropy shows a decrease - these compound events are easier to predict.

### 5.2  Projected changes in the near future

In a second step we calculate the change signal between 1971-2000 and 2021-2050 for all members of the CCLM-ensemble. An additional information of interest for the interpretation of the results is the change in the number of compound extreme days. The number of univariate extreme days are kept constant when partitioning the data (see Sect. 2.3) but the combination can change. The climate change signal is calculated separately for each ensemble member and then the mean climate change signal (bar in the following plots) as well as the interquartile range (marked by the whiskers) of the individual change signals are calculated and pictured. The number of compound cold and wet extreme days increases in all regions except region 5 (Russia) between the two time periods 1971-2000 and 2021-2050 and the number of compound extreme days differs between the regions. Regions 1 and 6 (Spain and Bulgaria) show the highest number of compound extreme events. (see Fig. 8). The ensemble mean values of the descriptors for cold and wet extremes in winter are shown in Fig. 10, whiskers give the interquartile range. The significance of the change signal was calculated using the nonparametric MannWhitneyWilcoxon test which tests for a difference in location of the values of the ensemble for the two different time periods. The p-values are shown below the bars in the respective figures. About one third of these p-values are smaller than 0.5, thus significant at the 5% significance level, e.g. region 5 (Russia) shows a significant change signal for the persistence and some changes are significant at the 10 % or 20% significance level (p-value $\leq 0.1$ or $\leq 0.2$). Nevertheless we follow von Storch and Zwiers (2013), who question hypotheses testing on future climate ensembles and instead propose to better use "*a simple descriptive approach for characterizing the information in an ensemble of scenarios*". Being conscious about the difficulties which may arise during hypotheses testing, we look at the ensemble spread in form of the interquartile range to assess the robustness of the results, and consult the significance test to support our findings. In many cases, the majority of ensemble members show a change signal in the same direction and the change signal is of a similar order of magnitude as the observed past changes in the preceding section (Figs. 6 and 7). In addition, a comparison to the results of the error estimation using FT-surrogate time series (Tab. 2) yields that the changes are higher then the estimated error. Therefore we conclude that

future changes of the succession of cold and wet extremes in winter in some regions in Europe can be expected. These changes are, for the significant cases, larger than $50\,\%$ for the persistence, larger than $20\,\%$ for the recurrence time and larger than $5\,\%$ for the entropy. Regarding the findings from our sensitivity analysis (Sect. 3.2) such changes are larger than the natural variability of the descriptors, which hence can be ruled out as the cause. Further, the sensitivity study has shown that such

changes in the past occurred concurrently with a strong increase in $CO_2$ emissions. As explained in Sect. 2.3, the only difference between the model runs for the periods 1971-2001 and 2021-2050 is the $CO_2$ forcing, thus the most probable reason for these changes in the future is the increase in $CO_2$ emissions.

Fig. 10 reveals three regions which seem to be particularly susceptible to changes of the dynam-
ics / succession, namely regions 2 (France), 3 (Germany) and 5 (Russia). The persistence changes for all regions and cold and wet episodes are likely to be of longer duration in the future. In regions 2 and 3 (France and Germany) the recurrence time decreases. The consequences of these changes are that these regions will probably experience more and longer cold and wet events in winter. Further-more, these are less predictable (increase of entropy). The situation is different for region 5 (Russia),
here the duration of cold and wet periods probably increases as well, but the number of events stays constant. Thus the system resides for longer times in the non-extreme states (increase in recurrence time).

The change in number of compound hot and dry extreme days is depicted in Fig. 9. Here, the number of compound extreme days varies with the region (although the number of univariate ex-
tremes are kept the same). Region 1 (Spain) shows a relatively low number of compound hot and dry days (note: all extremes in this paper are relative), regions 5 and 6 (Russia and Bulgaria) have a high number and also the highest decrease between the two time periods. Except for region 3 (Germany), which shows a slight increase, the number of compound extremes decreases in all regions. However, the change is generally small, $< 10\,\%$. Thus, the observed changes of the descriptors can mostly
be attributed to the change in the dynamics and not to a change in the numbers of events, except maybe for regions 4 and 5 (Russia and Bulgaria). The change signal of the descriptors is pictured in Fig. 11. Two regions are most probably susceptible to changes in the dynamics of the hot and dry state, namely regions 1 (Spain) and 6 (Bulgaria). Region 1 shows a small increase in persistence and a quite strong increase in recurrence time (in the order of $20\,\%$) of the hot and dry state, the entropy
does not change. The hot and dry periods get longer but less frequent. Regarding again the sensitivity study (Sect. 3.2) it can be seen that a change of $20\,\%$ of the recurrence time in summer (JJA) is at least twice as large as the variability of the recurrence time (about $10\,\%$) from 1900 to 2015 and con-stitutes a fairly large jump. The situation for region 6 is similar to that of region 1, with an increase in persistence and recurrence time and only a very small change in entropy. In addition, Region 3
(Germany) shows an increase in persistence and a decrease in entropy. This means the episodes will

be longer and more regular whereas in region 5 (Russia) the persistence slightly decreases and the recurrence time increases. This implies changes towards shorter and less frequent events.

## 6   Conclusions and Outlook

The changing climate leads to a change in extreme weather, which comprises several aspects like
frequency, duration, intensity etc. On top of these rather linear changes, modifications of the complex succession of extremes can be expected. However, information on the succession or dynamical behavior of climate extremes is rare. Therefore, to extract such information from climate time series we applied a Markov chain analysis on compound extremes, namely cold and wet in winter and hot and dry in summer. We have shown that our climate model ensemble is able to reproduce past dy-
namics of compound extremes fairly well within acceptable uncertainties. Thus, we have reasonable confidence in the future simulations of this model ensemble. We identified three regions in Europe, which are probably susceptible to a future change in the succession and dynamical behavior of cold and wet extremes in winter. In region 5 (Russia) we detected an increase of the persistence and recurrence time, which means that the probability of staying in the cold and wet state from one day to the
next will increase, but the system will take longer to approach this state again. In regions 2 (France) and 3 (Germany), cold and wet episodes become both longer and more frequent. The entropy in these regions also increases in the future, which is counterintuitive, because one would expect that an increase in persistence is related to a decrease in entropy (cf. Eqs. 5 and 7). However, since the entropy (Eqs. 7) does not only consider the compound extreme state but also transitions from this
state to the normal state and univariate extreme states, complex interactions can be extracted with the entropy. The impacts of these calculated changes are beyond the scope of this study, and it can only be speculated about possible effects. One could imagine that longer and less predictable cold and wet periods could lead to larger snow chaos regarding traffic and other human life, especially in regions which already experience extreme cold temperatures in winter. Again, these findings suggest
that a reordering of the succession of compound extremes could be happening on top of the observed linear changes, as e.g. the temperature increase.

For hot and dry states in summer, the Markov method identified two regions where changes are probable, Spain and Bulgaria. The persistence and recurrence time in regions 1 and 6 (Spain and Bulgaria) both increase in the future, which means that the system resides longer in the extreme
state. The entropy does not change significantly. Any reordering of the succession of extremes has an impact. For instance such changes could be harmful for the local agriculture, because, as explained above, these dynamic changes would occur on top of the known linear increase of e.g. temperatures. Interestingly, in region 6 (Bulgaria) the absolute number of compound hot and dry extremes (Fig.9) decreases in the future, but the extreme periods become longer. The changes for region 3 (Russia)
are small but indicate that the region in Russia near Moscow will be less susceptible to dynamical

changes of the succession of compound extremes and will additionally experience less compound extremes in the near future.

A number of studies have shown an influence of atmospheric drivers (mostly NAO) and atmospheric blocking patterns on summer as well as winter temperature extremes and generally the temperature variability in Europe (e.g. Photiadou et al., 2014; Sillmann and Croci-Maspoli, 2009). Although the extremes analyzed in these studies were mostly of absolute nature, an analysis of the influence of the same factors on the relative extremes studied in this paper would be very interesting. Using a similar methodology as described in this paper to calculated persistence, recurrence and entropy of time series of e.g. the NAO index in a certain regime could be linked to the descriptors of the compound extreme events.

Areas to apply this method are manifold. Besides the analysis of different dynamical behavior varying on the region and extreme considered, it can be used as a model validation tool. As extremes and especially compound extremes are an important quantity that we want to assess with climate model data, it is necessary for the models to capture the dynamical behavior of these extreme events. As shown in this paper, the models can also project changes of the future dynamical behavior which is an interesting supplementary information to changes in mean and variability. An example where this could be useful is the decision whether to apply simple or more sophisticated bias correction techniques.

Follow up studies using simulations of other regional climate models and regional climate ensembles for time periods further in the future (e.g. ENSEMBLES, http://ensembles-eu.metoffice.com/, or CORDEX, http://www.euro-cordex.net/, data for the end of the century) would be interesting. For one, this would allow an analysis of whether or not there are significant differences depending on the regional climate model used. In addition, data for the end of the 21st century is available where changes in the descriptors could possibly be larger because the influence of the $CO_2$ forcing plays a more important role. In this sense, the Markov chain analysis could be useful to identify possible future regime shifts (Scheffer and Carpenter, 2003; Scheffer et al., 2009). Of further interest is an analysis of the dynamical behavior of the driving GCMs as well as the ERA-40 reanalysis dataset since for parts the ERA-40 driven CCLM model runs performed worse in comparison to observations than the CCLM ensemble. This leads to the question whether or not the CCLM model runs compensate for errors in the driving GCMs and are right for the wrong reasons. Comparison of the E-OBS dataset to other regionally defined datasets would also be helpful to evaluate the observational data.

*Acknowledgements.* We acknowledge the E-OBS dataset from the EU-FP6 project ENSEMBLES (http://ensembles-eu.metoffice.com) and the data providers in the ECA&D project (http://www.ecad.eu). Figs.1 and 3 were made using the GMT (Generic Mapping Tools) web-application www.piece-of-earth.net. All other graphics were made using R (R Development Core Team, 2008). We also thank P.Berg and R.Sasse for their contributions to

the CCLM-Ensemble. The authors thank members of the IMAGE and RCR sections at NCAR for the fruitful discussions and the three anonymous referees for their helpful comments and suggestions.

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

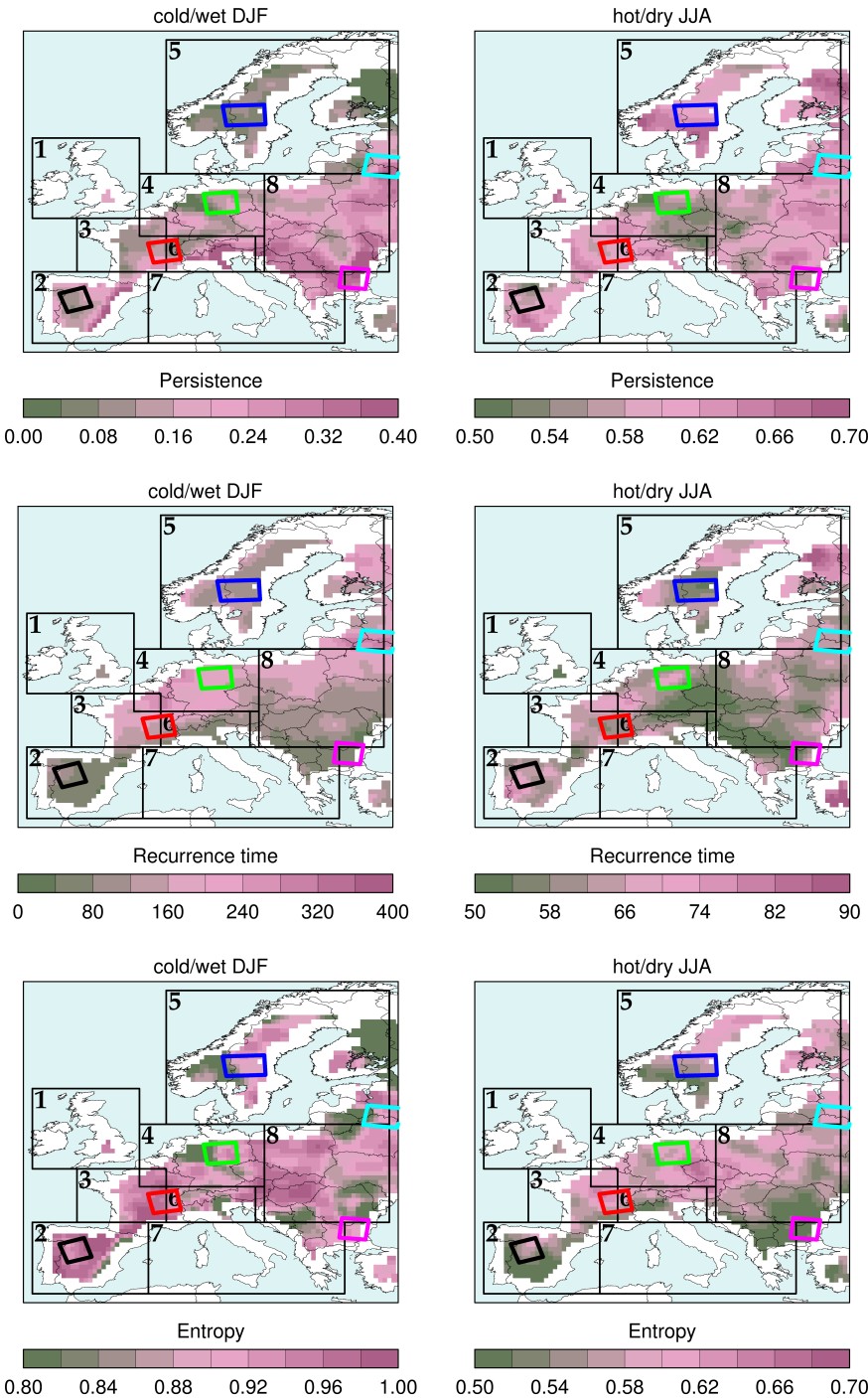

**Figure 1.** E-OBS descriptors for the reference period (1971-2000). Left side: Descriptors for cold and wet extremes in winter (DJF) (Ta < 10th percentile and Pa > 75th percentile). Right side: Descriptors for hot and dry extremes in Summer (JJA) (Ta > 90th percentile and EDI < 25th percentile). Descriptors were calculated for a moving window over 9 gridpoints and values assigned to the center grid point (see text). Boxes show the Prudence Regions (http://ensemblesrt3.dmi.dk/quicklook/regions.html)

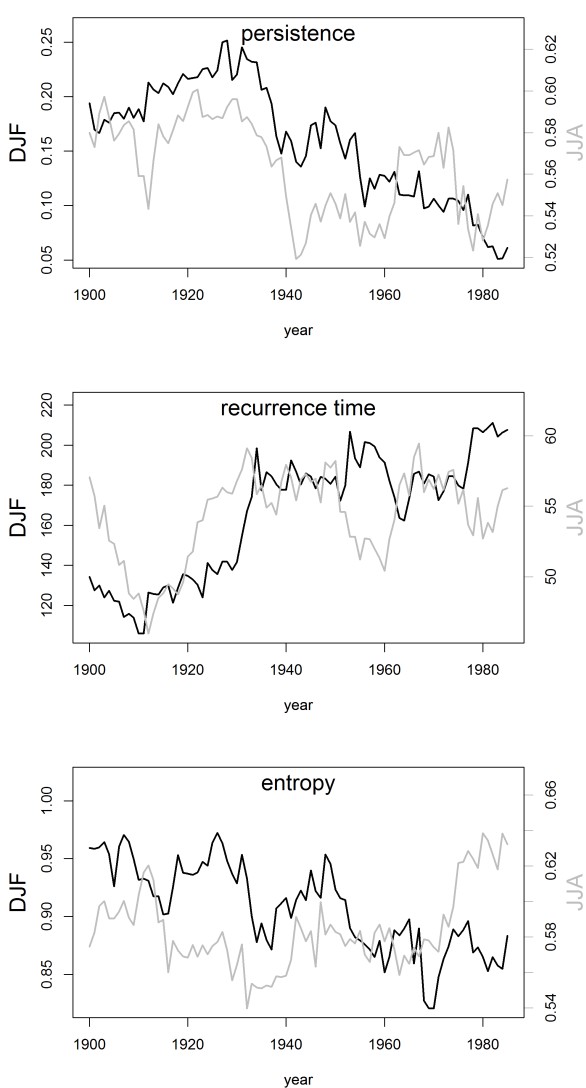

**Figure 2.** Descriptors for ECA& D station data for running windows over 30-years (values are assigned to the first year of the 30-year time period.) from 1900-2015. Black curve: cold and wet extremes in winter (DJF) (Ta < 10th percentile and Pa > 75th percentile). Gray lines: hot and dry extremes in Summer (JJA) (Ta > 90th percentile and EDI < 25th percentile)

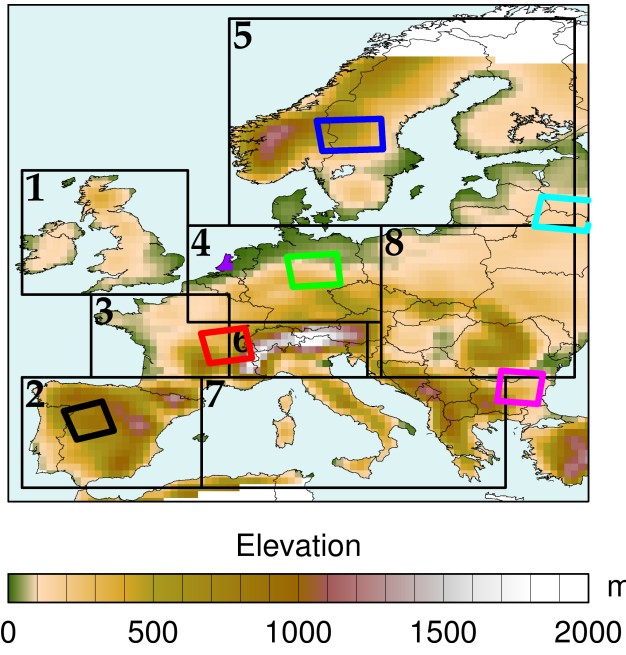

**Figure 3.** Elevation of the CCLM 50 km Model domain [m]. Boxes mark the six investigation areas, 1:Spain (black), 2:France (red), 3:Germany (green), 4:Scandinavia (blue), 5:Russia (cyan) and 6:Bulgaria (magenta).

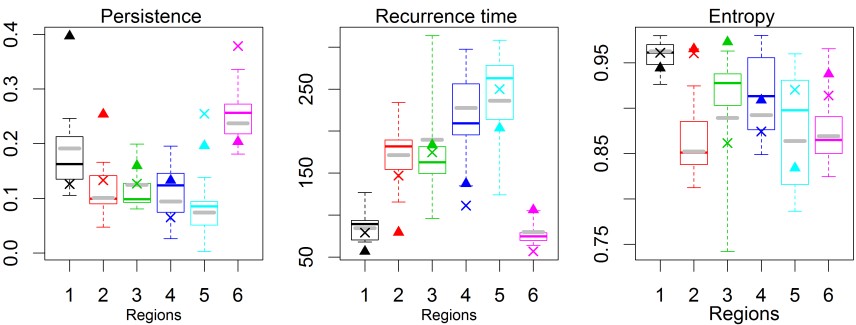

**Figure 4.** Descriptors for cold and wet extremes in winter (DJF) (Ta < 10th percentile and Pa > 75th percentile) in the reference period 1971-2000 for the 6 investigation areas. Box plots of the CCLM ensemble: box = ensemble median and interquartile range, whiskers = ensemble minimum/maximum, gray bars: ensemble mean, triangles :reanalysis driven CCLM runs, crosses: E-OBS observations. The coloring corresponds to the regions in Fig. 3.

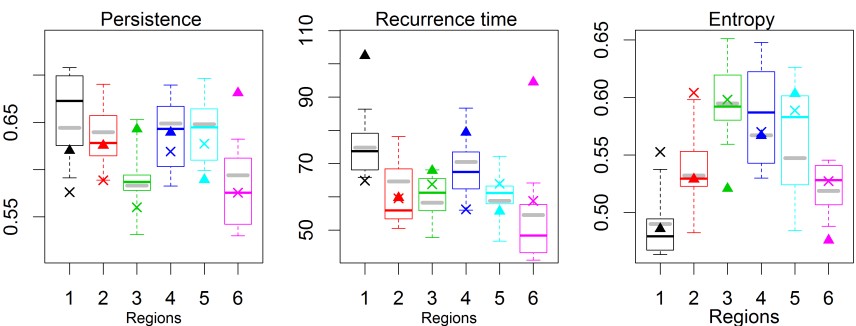

**Figure 5.** Descriptors for hot and dry extremes in Summer (JJA) (Ta > 90th percentile and EDI < 25th percentile) in the reference period 1971-2000 for the 6 investigation areas. Box plots of the CCLM ensemble: box = ensemble median and interquartile range, whiskers = ensemble minimum/maximum, gray bars: ensemble mean, triangles :reanalysis driven CCLM runs, crosses: E-OBS observations. The coloring corresponds to the regions in Fig. 3.

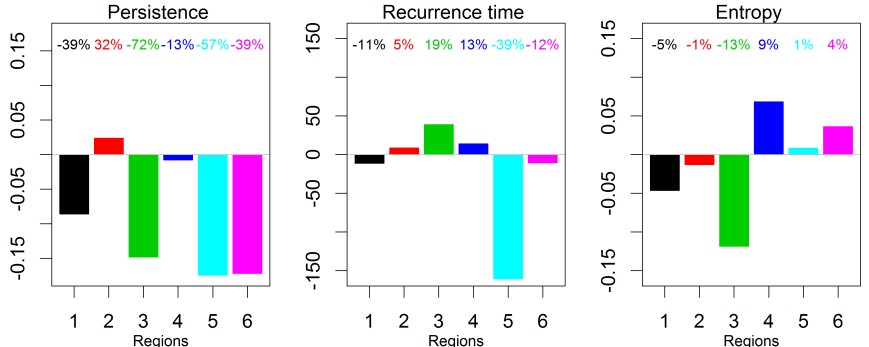

**Figure 6.** Change signal of descriptors for E-OBS observations: Cold and wet extremes in winter (DJF) (Ta < 10th percentile and Pa > 75th percentile). Changes between the time periods 1951-1980 and 1981-2010. Percentages denote the relative change. The coloring corresponds to the regions in Fig. 3.

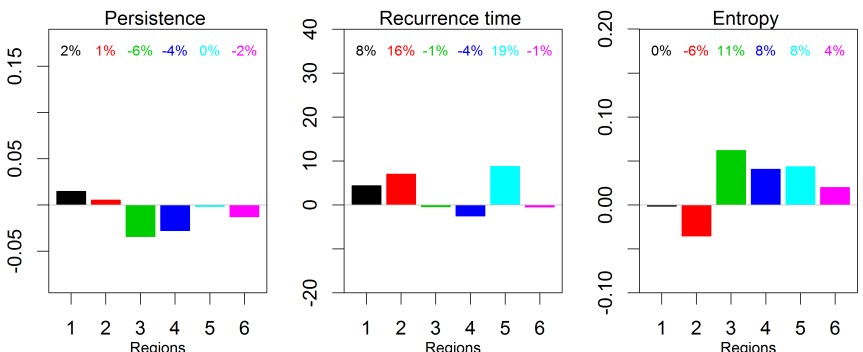

**Figure 7.** Change signal of descriptors for E-OBS observations: Hot and dry extremes in Summer (JJA) (Ta > 90th percentile and EDI < 25th percentile). Changes between the time periods 1951-1980 and 1981-2010. Percentages denote the relative change. The coloring corresponds to the regions in Fig. 3.

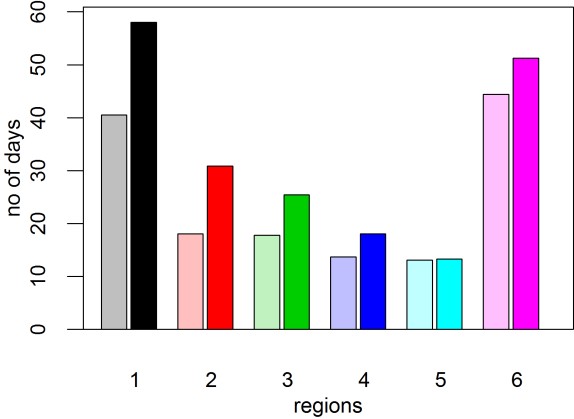

**Figure 8.** Number of compound cold and wet extremes in winter (DJF) (Ta < 10th percentile and Pa > 75th percentile) , 1971-2000 (light colors) and 2021-2050 (dark colors), ensemble mean. The coloring corresponds to the regions in Fig. 3.

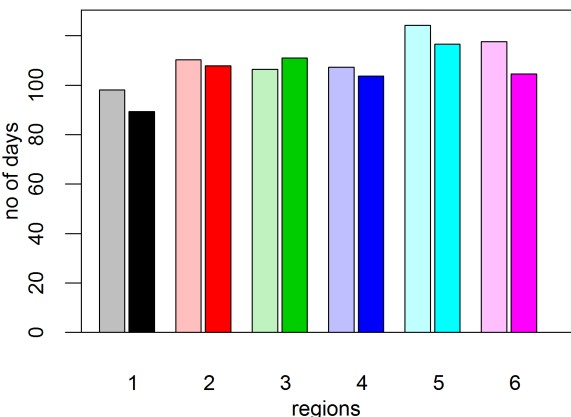

**Figure 9.** Number of compound hot and dry extremes in summer (JJA) (Ta > 90th percentile and EDI < 25th percentile), 1971-2000 (light colors) and 2021-2050 (dark colors), ensemble mean. The coloring corresponds to the regions in Fig. 3.

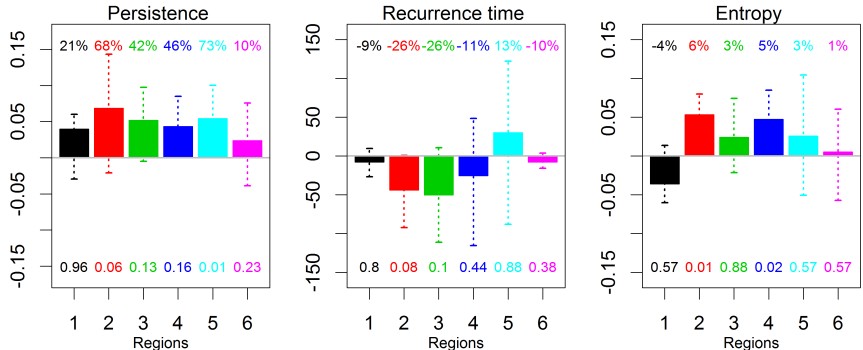

**Figure 10.** Climate change signal of descriptors for cold and wet extremes in winter (DJF) (Ta < 10th percentile and Pa > 75th percentile). Changes between the time periods 1971-2000 and 2021-2050. Bars show the ensemble mean (of the individual change signals), whiskers the 75th and 25th quantile, respectively. Percentages above the bars denote the relative change of the ensemble mean, the numbers below the p-value. The coloring corresponds to the regions in Fig. 3.

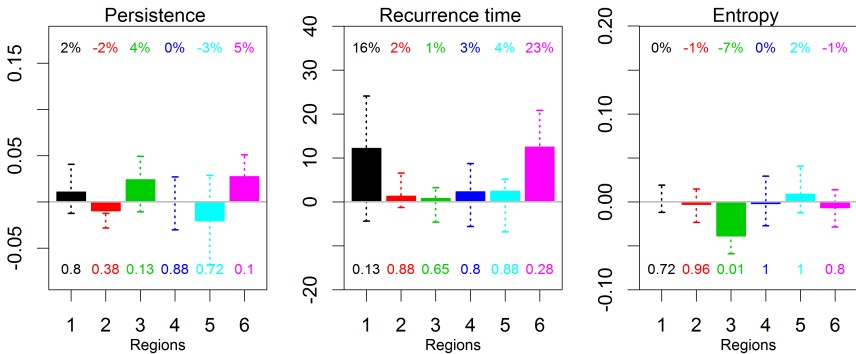

**Figure 11.** Climate change signal of descriptors for hot and dry extremes in Summer (JJA) (Ta > 90th percentile and EDI < 25th percentile). Changes between the time periods 1971-2000 and 2021-2050. Bars show the ensemble mean (of the individual change signals), whiskers the 75th and 25th quantile, respectively. Percentages above the bars denote the relative change of the ensemble mean, the numbers below the p-value. The coloring corresponds to the regions in Fig. 3.