# Peer review of "Compound extremes in a changing climate - a Markov Chain approach"

_Nonlinear Processes in Geophysics, 2015_

## Referee Comment (RC1) · Anonymous Referee #1 · 23 Mar 2016

General: This is an interesting article examining changes in the occurrence and likely impacts of compound extremes. It addresses a problem which is not well represented in the literature, namely how frequently extreme events occur in close succession and whether there will likely be changes in the future. The method employs a novel method using a first order markov chain to describe the persistence, recurrence time and predictability of repeated extreme events. The paper is clearly written, subject to some minor corrections below, and well presented.

Specific: Section 3 discussion of results (approx L250 on), it would be good to see some comparison with other research on the persistence of extremes in different regions and possible causes. e.g. Sillmann & Croci-Maspoli 2009, Furrer et al 2010, Photiadou et al 2014. Furrer, E.M., R.W. Katz, M.D. Walter, and R. Furrer, 2010: "Statistical modeling of hot spells and heat waves." Climate Research, 43, 191-205

Photiadou, C., Jones, M., Keellings, D., Dewes, C., 2014. Modeling European hot spells using extreme value analysis. Clim. Res. 58, 193–207. doi:10.3354/cr01191
Sillmann, J., Croci-Maspoli, M., 2009. Present and future atmospheric blocking and its impact on European mean and extreme climate. Geophys. Res. Lett. 36, L10702. doi:10.1029/2009GL038259

Similarly a sentence or two comparing the reliability of different models and observations would be good - e.g. CFSR and ERA-40 can be very different. This could be in the data section.

Did you test the significance of the changes in the reference period as well as the future? How did you account for uncertainty in the results?

L338 note about relative extremes - This should really be mentioned in the method section along with how you selected the extremes (e.g. thresholds, and at which level). Possibly a table of extremes would be informative for comparison?

Technical corrections: L3 "the number of occurrences" L9 types L11 replace "which are" with "including" L12 rogue comma before fullstop. L26 occurrences L36 changes in the number of L46 should this be chaotic attractor? L107 please put into present tense to match the rest of the text. L115 ditto L145 unnecessary comma at start of line. L180 "number of states" L189&192 "Thus in the sense of successive compound ..." L216 should this be per 100 days? L245 maybe say very rare? There are a lot of extremes in that sentence. L273 highest persistence is

Figure 9 caption rogue fullstop before Percentages.
* * *

---

## Referee Comment (RC2) · Anonymous Referee #2 · 23 Mar 2016

Review of Manuscript # npg-2015-74

Compound extremes in a changing climate - a Markov Chain approach

by K. Sedlmeier et al.

The manuscript presents an interesting idea of studying compound extremes using the Markov chain model. The authors construct 4-state models by defining 2 states (extreme, nonextreme) for two variables. Thus they can follow combined extremes hot+dry in summer and cold+wet in winter. The dynamics of these compound extremes are characterized by 3 quantitative descriptors based on the constructed Markov models: persistence, recurrence time, entropy. Then they show how these descriptors change in two segments of experimental data and in model simulations.

The descriptors in this context are new and original. This is, of course, expected for a novel publication, however, one should demonstrate that new descriptors reasonably reflect underlying physical mechanisms. Before using any new measure for characterization of ongoing and expected climate change, one should investigate their variability in natural conditions. The authors use the gridded E-OBS dataset, however, they unfortunately chose just a few gridpoints in six different areas. It is a pity, since the E-OBS dataset gives an excellent opportunity to study spatial variability of any descriptor which has an ambition to characterize the temporal evolution of a physical quantity attributed to each gridpoint. I think the model is reasonably simple to compute full coverage for Europe for all three descriptors and map them. The simple visual evaluation would indicate if the descriptors reasonably reflects physical reality in the case the maps show interpretable smoothly changing patterns. Or, if the maps show just a colored grains or a sort of Pollock's paintings, than there is a problem with the descriptor and its connections to physical reality.

While E-OBS dataset can be used to test spatial variability, ECA&D station dataset offers a number of long-term records in which temporal variability can be tested. So one can relate the change of the introduced descriptor due to climate change to their changes due to natural variability in preindustrial era.

Real long-term records would reflect natural variability due to natural nonstationarity. One can test numerical variability of the descriptors by constructing appropriate surrogate data. E.g., FT surrogate data generation averages dynamics over whole record randomized, so one can get ranges for random variability of the descriptors in a stationary data.

Only after understanding how any new descriptor behaves, it can be used to quantify an evolution or change of interest.

Technical remarks:

P. 9, last para

....Fig. 4) all regions except Bulgaria...

Should not it be France?

p 10–, 4.2

The statistical treatment should be described in more details: Differences of the ensemble means are plotted, i.e. one get the mean and percentiles for each ensemble, then the difference of means is clearly defined, but what are the percentiles?

Is this an appropriate way to evaluate the significance of changes?
* * *

---

## Referee Comment (RC3) · Anonymous Referee #3 · 1 Apr 2016

The authors apply an idea which was previously introduced by Mieruch et al. (2010), studying the sensitivity of Markov chain quantitative descriptors (persistence, recurrence time and Shannon entropy) for compound events (extremes of temperature and precipitation and/or EDI) with respect to: a) different European regions, b) different recent climate-normal periods of 30 years length and c) different periods comparing the present climate and a future climatic scenario. The ensemble variability of descriptors is obtained from ensemble model downscaled simulations.

The proposed method is an interesting approach, among many other possible diagnostics for studying the dynamics and trends of joint (compound) extremes of different physical properties.

[Figure]

Some points must be addressed:

1 – The authors should refer other approaches like the geostatistical analysis of spatially distributed extremes (Neves 2015). That is important because extremes have themselves some spatial organization.

2 – There is no clear justification for the choice of the 6 box-regions and their size (6x6 grid points). Why they are representative of the PRUDENCE regions? Some minimal study about the spatial robustness of the Markov diagnostics should be presented. For example, does the results keep similar or change substantially when contiguous boxes are considered? The ideal should be to present maps of the diagnostics throughout Europe.

Minor corrections

3 - In the entropy definition H (eq. 7), log(1/m) must be replaced by log(m) so that H equals 1 for a random system without memory (all probabilities pij=1/m).

4 - Line 189: Authors claim that H between 0 and 1 is an identification of deterministic chaotic behavior. However that condition is necessary but not a sufficient condition for chaos. Authors shall carefully rephrase the paper by taking that into account.

5 - Line 197: Authors say 'The reason for this is that the $CO_2$ forcing is the only difference...'. In fact, decadal variability is also likely. That sentence must be weakened by replacing 'the only' by 'the main difference beyond the natural decadal variability'.

6 - Eq. 8 – explain the meaning of the bar and subscripts rm.

7 - Line 234: Droughts may have different time scales from months to years. That is the reason for defining the SPI (Standard Precipitation index) (McKee et al. 1993). The presented EDI is appropriate for annual scaled droughts. Add this comment to the text. Moreover the EDI has its own annual cycle since the precipitation weights contributing to EDI are larger near the Julian day d. Does the annual cycle of EDI was removed?

8 - L235-238 Does temperature anomalies (Ta) and precipitation anomalies (Pa) refer to daily Ta and daily Pa with respect to the respective annual cycle. Please clarify. Add a sentence about the number of categories of the Markov chain and what categories of the compound attractor were considered? I suppose that authors have considered 2 parameters with a partition of 2 categories each. Confirm that at this stage for the sake of the paper understanding.

9 - Fig. 3 In the recurrence plot I cannot see the black triangle for region 1.

10 - Fig. 4 In the caption, descriptors' changes refer to changes in the period 1981-2010 with respect to 1951-1980? Rewrite it in a clearer way.

References:

McKee, T.B., Doeskin, N.J., Kleist, J., 1993. The relationship of drought frequency and duration to time scales. Eighth Conf. on Applied Climatology, American Meteorological Society, 179–184.

Neves, M.M. 2015. Geostatistical Analysis in Extremes: An Overview. Mathematics of Energy and Climate Change. Volume 2 of the series CIM Series in Mathematical Sciences pp 229-245. DOI 10.1007/978-3-319-16121-1_10

---

## Author Comment (AC1) · 29 Jun 2016

**Answer to comment of referee #3**

**Compound extremes in a changing climate - a Markov Chain approach**

**K.Sedlmeier, S. Mieruch, G. Schädler and C. Kottmeier**

Dear referee,

Thank you for your detailed review of the paper. In the following, you can find our answers to your comments which are written in red text color.

**1 General comments**

**The authors should refer other approaches like the geostatistical analysis of spatially distributed extremes (Neves 2015). That is important because extremes have themselves some spatial organization.**

That is an interesting comment and we will refer to this in the introduction of the revised version.

**There is no clear justification for the choice of the 6 box-regions and their size (6x6 grid points). Why they are representative of the PRUDENCE regions? Some minimal study about the spatial robustness of the Markov diagnostics should be presented. For example, does the results keep similar or change substantially when contiguous boxes are considered? The ideal should be to present maps of the diagnostics throughout Europe.**

We thank the referee for that comment, because indeed, we have performed tests on the robustness of the Markov descriptors, which are the basis for the decision to use the actual 6 box-regions. The crucial point, why we have used the 6 box-regions is to achieve that each region contains the exact same amount of grid points / data points. This is of utmost importance for the comparison of the regions, otherwise, if the regions have been chosen with differing sizes no consistent comparison would be possible due to the fact that the Markov descriptors depend on the underlying sample size of the used data. To account for the spatial robustness we calculated the Markov descriptors for every grid point in Europe and visualised the results on a maps. From these maps we have seen that the Markov descriptors vary in general not strongly within the prudence regions. Accordingly we have chosen the 6 box-regions within the Prudence regions, which are representative for the respective region, based on the results of the grid point maps. In the revised version we will include the maps showing the grid point results (see Fig. 1)

**2   Minor corrections**

**In the entropy definition H (eq. 7), log(1/m) must be replaced by log(m) so that H equals 1 for a random system without memory (all probabilities pij=1/m).**

Thank you for the comment but our definition corresponds to those of other papers (see eg. Hill et al., 2004). Maybe you have missed the - sign at the beginning or the "'/"' sign in the equation (log(1/m) is the same as -log(m))? By using log(m) we would get negative entropies with our formula.

**Line 189: Authors claim that H between 0 and 1 is an identification of deterministic chaotic behavior. However that condition is necessary but not a sufficient condition for chaos. Authors shall carefully rephrase the paper by taking that into account.**

We agree with this comment and thank you for the notice. We will rephrase the follwing sentence:
*The dynamics of complex chaotic systems lie in between these limits, thus the entropy can be used to identify and characterize complex dynamics like deterministic chaos, which is not possible with standard linear methods*
by
*The dynamics of complex chaotic systems lie in between these limits, thus the entropy can give a hint to underlying complex dynamics like deterministic chaos, which is not possible with standard linear methods. To really test for deterministic choas other methods, based on state space reconstruction (e.g. estimating the correlation dimension, Lyapunov exponents etc.) to find strange attractors, are more suitable.*
and rephrase references to the chaotic behavior accordingly throughout the revised version of the paper.

**Line 197: Authors say The reason for this is that the CO2 forcing is the only difference. . .. In fact, decadal variability is also likely. That sentence must be weakened by replacing the only by the main difference beyond the natural decadal variability.**

No, because the crucial point is that this sentence (line 197) refers to the **model runs** (cf. line 198). The decadal variability of the model is not intrinsically changing with time. The only difference between the model runs in the past and in the future is the $CO_2$ forcing. Thus, changes of the decadal variability are of course possible, but the only reason is a changing $CO_2$ forcing.

**Eq. 8 explain the meaning of the bar and subscripts rm.**

Yes, we will do so and also include a more detailed explanation of the EDI in the revised version (also see next comment). The bar in equation 8 stands for the climatological mean - $\overline{EP_{d,rm}}$ refers to the climatological mean state of EP corresponding to day d, where the climatological mean is calculated by a running mean of rm days over the 30 years of the respective time period.

**66  Line 234: Droughts may have different time scales from months to years. That is**
**67  the reason for defining the SPI (Standard Precipitation index) (McKee et al. 1993).**
**68  The presented EDI is appropriate for annual scaled droughts. Add this comment to**
**69  the text. Moreover the EDI has its own annual cycle since the precipitation weights**
**70  contributing to EDI are larger near the Julian day d. Does the annual cycle of EDI**
**71  was removed?**

72  The EDI does not have an annual cycle as this is intrinsically removed by the method
73  (e.g. http://atmos.pknu.ac.kr/~intra2/eng.calculation.htm). In the equation:

$$EDI_d = \frac{EP_d - \overline{EP_{d,rm}}}{\sigma\left(EP - \overline{EP}\right)_d} \tag{1}$$

74  $\left(\overline{EP}\right)_d$ refers to the climatological mean state of EP and is calculated for each day as the 5day
75  running mean over the 30years of the respective time period. Thus, by subtracting $\left(\overline{EP}\right)_d$ from
76  EP, the annual cycle is removed. We are sorry that this did not become clear and will include a
77  better explanation of the EDI in the methods section of the revised version and clearly state that
78  the annual cycle is removed by the method.

**79  L235-238 Does temperature anomalies (Ta) and precipitation anomalies (Pa) refer to**
**80  daily Ta and daily Pa with respect to the respective annual cycle. Please clarify. Add**
**81  a sentence about the number of categories of the Markov chain and what categories**
**82  of the compound attractor were considered? I suppose that authors have considered**
**83  2 parameters with a partition of 2 categories each. Confirm that at this stage for the**
**84  sake of the paper understanding.**

85  Yes, the Ta and Pa refer to daily temperature and precipitation anomalies with respect to the annual
86  cycle. And we have considered 2 parameteres with a partition of 2 categories each which we then
87  combined to a 4 state symbolic sequence. We will add a more detailed description of the anomaly
88  calculation and partitioning in the methods section.

**89  Fig. 3 In the recurrence plot I cannot see the black triangle for region 1.**

90  thank you for the notice, we will change that.

**91  Fig. 4 In the caption, descriptors changes refer to changes in the period 1981- 2010**
**92  with respect to 1951-1980? Rewrite it in a clearer way.**

93  We will replace *1951-1980 vs 1981-2010.* by *changes between the time periods 1951-1980 and*
94  *1981-2010*

[Figure]

Figure 1: E-Obs descriptors for the reference period (1971-2000). Left side: Descriptors for cold and wet extremes in winter (DJF) (Ta < 10th percentile and Pa > 75th percentile). Right side: Descriptors for hot and dry extremes in Summer (JJA) (Ta > 95th percentile and EDI < 25th percentile). Descriptors were calculated for a moving window over 9 gridpoints and values assigned to the center grid point. Boxes show the Prudence Regions (http://ensemblesrt3.dmi.dk/quick-look/regions.html).

**References**

Hill, M., J. Witman, and H. Caswell, 2004: Markov chain analysis of succession in a rocky subtidal community. *Am. Nat.*, **164 (2)**, E46–E61, doi:10.1086/422340.

---

## Author Comment (AC2) · 30 Jun 2016

**Answer to comment of referee #1**

**Compound extremes in a changing climate - a Markov Chain approach**

K.Sedlmeier, S. Mieruch, G. Schädler and C. Kottmeier

Dear referee,

Thank you for your detailed review of the paper. In the following, you can find our answers to your comments which are written in red text color.

**1   Specific comments**

**Section 3 discussion of results (approx L250 on), it would be good to see some comparison with other research on the persistence of extremes in different regions and possible causes. e.g. Sillmann & Croci-Maspoli 2009, Furrer et al 2010, Photiadou et al 2014. Furrer, E.M., R.W. Katz, M.D. Walter, and R. Furrer, 2010: "Statistical modeling of hot spells and heat waves." Climate Research, 43, 191-205 Photiadou, C., Jones, M., Keellings, D., Dewes, C., 2014. Modeling European hot spells using extreme value analysis. Clim. Res. 58, 193–207. doi:10.3354/cr01191 Sillmann, J., Croci-Maspoli, M., 2009. Present and future atmospheric blocking and its impact on European mean and extreme climate. Geophys. Res. Lett. 36, L10702. doi:10.1029/2009GL038259**

This is a good point and we will include a more thourough comparison with other research on the persistence of extremes in the discussion section of the revised version.

**Similarly a sentence or two comparing the reliability of different models and observations would be good - e.g. CFSR and ERA-40 can be very different. This could be in the data section.**

Thank you for this comment, we will include this in the data section (Section 2.1. in the original manuscript). An additional interesting application of the method is also to detect differences in observational datasets and models concerning the dynamical behavior of extreme events.

**Did you test the significance of the changes in the reference period as well as the future? How did you account for uncertainty in the results?**

Regarding the **uncertainty**, we took advantage of the applied ensemble approach. In Fig. 2 (of the original manuscript) we show the results of the ensemble for the reference period, where we use a box plot for the ensemble: box = ensemble median and interquartile range, whiskers = ensemble minimum/maximum, gray bars: ensemble mean. This information is given in the text caption, to make it clear, we will include it in the text under Sect. 4 (of the original manuscript).

Similar box plots accounting for the ensemble uncertainty have been used in Figs. 3, 8, 9 (where for the change signal, the changes were calculated for each ensemble member individually and then displayed in the same manner). As can also be seen from the figures we did not account for the uncertainties in the observational E-OBS dataset and consider the observations approximately as the truth. Nevertheless we will include an additional section in the revised version where we calculated the error of the descriptors by a FT-resampling algorithm. For this we used the MIAAFT algorithm (Venema et al., 2006) which in addition to preserving the original distribution of the data also preserves the auto and cross-correlation of the temperature and precipitation time series. 100 surrogate data sets for the 6 regions used throughout the paper were calculated for the E-Obs data set in the reference period (1971-2000) and their standard deviation taken as the error (by using the exact same regions the values are transferable to later chapter which would not be possible had we chosen a different number of data points). An overview of the errors can be seen in Tab. 1. In comparison to differences between regions and time periods, the error is small but we will include it in the discussions of Sect. 3 and 4. Regarding the **significance** we use the ensemble uncertainty, as mentioned above, and show in Sect. 4.2 that we use the nonparametric Mann-Whitney-Wilcoxon test for the change signal (Figs. 8, 9). The p-values are shown below the bars in the respective figures.

|        | DJF   |       |       | JJA   |       |       |
| ------ | ----- | ----- | ----- | ----- | ----- | ----- |
|        | P     | R     | E     | P     | R     | E     |
| reg1   | 0.010 | 1.701 | 0.004 | 0.007 | 1.183 | 0.009 |
| reg2   | 0.011 | 2.182 | 0.010 | 0.010 | 2.055 | 0.010 |
| reg3   | 0.010 | 2.563 | 0.005 | 0.009 | 0.923 | 0.007 |
| reg4   | 0.008 | 1.150 | 0.005 | 0.008 | 0.990 | 0.011 |
| reg5   | 0.010 | 2.45  | 0.010 | 0.008 | 1.103 | 0.009 |
| reg6   | 0.007 | 0.797 | 0.004 | 0.009 | 1.150 | 0.009 |

Table 1: Estimation of the error of the descriptors by using MIAAFT surrogates for winter (DJF) and summer(JJA) extremes. Values are calculated for the 6 regions of Fig. 1.

**L338 note about relative extremes - This should really be mentioned in the method section along with how you selected the extremes (e.g. thresholds, and at which level). Possibly a table of extremes would be informative for comparison?**

We mentioned the thresholds in a later section of the text:

- cold and wet in winter (DJF): temperature anomaly (Ta) < 10th percentile and precipitation anomaly (Pa) > 75th percentile)

- heat and drought in summer (JJA): Ta >95th percentile and EDI < 25th percentile

but it is a good idea and we will include the thresholds in the methods section.

**2   Technical corrections**

**L3 "the number of occurrences" L9 types L11 replace "which are" with "including" L12 rogue comma before fullstop. L26 occurrences L36 changes in the number of L46 should this be chaotic attractor? L107 please put into present tense to match the rest of the text. L115 ditto L145 unnecessary comma at start of line. L180 "number of states" L189 and 192 "Thus in the sense of successive compound ..." L216 should this be per 100 days? L245 maybe say very rare? There are a lot of extremes in that sentence. L273 highest persistence is Figure 9 caption rogue fullstop before Percentages.**

thank you for the correction of our english. In L216 1000days is correct because this number refers to the total number of days, not only the compound extreme states.

**References**

Venema, V., Meyer, S., García, S. G., Kniffka, A., Simmer, C., Crewell, S., Löhnert, U., Trautmann, T., and Macke, A.: Surrogate cloud fields generated with the iterative amplitude adapted Fourier transform algorithm, Tellus A, 58, 104–120, 2006.

---

## Author Comment (AC3) · 30 Jun 2016

**Answer to comment of referee #2**

**Compound extremes in a changing climate - a Markov Chain approach**

**K.Sedlmeier, S. Mieruch, G. Schädler and C. Kottmeier**

Dear referee,

Thank you for your detailed review of the paper. In the following, you can find our answers to your comments which are written in red text color.

**1   General comments**

In order to address your first comments, we will introduce a new section in the revised version called "Sensitivity analysis" where we address the spatial and natural variability and analyze the error by means of Fourier-Transform surrogate time series. Detailed comments can be found below.

**one should demonstrate that new descriptors reasonably reflect underlying physical mechanisms. Before using any new measure for characterization of ongoing and expected climate change, one should investigate their variability in natural conditions. The authors use the gridded E-OBS data set, however, they unfortunately chose just a few grid points in six different areas. It is a pity, since the E-OBS data set gives an excellent opportunity to study spatial variability of any descriptor which has an ambition to characterize the temporal evolution of a physical quantity attributed to each grid point. I think the model is reasonably simple to compute full coverage for Europe for all three descriptors and map them. The simple visual evaluation would indicate if the descriptors reasonably reflects physical reality in the case the maps show interpretable smoothly changing patterns. Or, if the maps show just a colored grains or a sort of Pollocks paintings, than there is a problem with the descriptor and its connections to physical reality.**

We thank the referee for that comment and totally agree that new descriptors must be tested for revealing a connection to physical reality. Indeed, we did these tests prior to our analysis, which were also the basis for choosing the regions discussed in this paper. We have calculated a full coverage for the descriptors averaging over 3x3 grid points for the whole area and these maps show interpretable smoothly changing patterns as you can see in Fig. 1. This figure will be included and discussed in the revised version of the paper in the newly introduced section.

As to Pollock's painting: a map like a Pollock's painting might not be achieved easily for the Markov descriptors. Pollock's paintings are not random and not noise, rather they are in between determinism and noise, they are fractal (Taylor et al., 2007, , and citations therein). Thus, due to

their fractal geometry they have deep underlying mechanisms in common with natural patterns and hence also with our atmospheric time series.

**While E-OBS data set can be used to test spatial variability, ECA&D station data set offers a number of long-term records in which temporal variability can be tested. So one can relate the change of the introduced descriptor due to climate change to their changes due to natural variability in preindustrial era. Real long-term records would reflect natural variability due to natural nonstationarity.**

This is a good suggestion. Unfortunately, there is only one station with a **continuous** (without missing values) temperature and precipitation record (starting in 1887) available from the ECA&D data set. Further, only a few stations within Germany have available **continuous** time series starting in 1900. Nevertheless we calculated the descriptors for a combined time series of the available 7 stations in Germany for running windows of 30 years starting in 1900. The combination of the time series is necessary in order to fulfill the stationarity criteria explained in Section 2.3. of our original manuscript (non zero entries of the transition probability matrix and stationarity of the time series). It is important to note that we removed all linear trends for each 30 year section seperately as it has been done in the rest of the paper. The resulting time series of the descriptors are shown in Fig. 2 for both winter (black) and summer (gray) extremes. These results will be included in the "Sensitivity analysis" section in the revised version of the paper. The stations used will be listed in the data section. Especially for the persistence and recurrence time, a clear shift is visible between 1930 and 1950. This time range is not preindustrial, but the crucial point is that the observed shift coincides with a globally oberved shift in the increase in CO2 around 1950 (http://www.ldeo.columbia.edu/~spk/Research/AnthropogenicCarbon/images/ddic_uptake_hist.png). Thus from this finding we observe two main points:

1. The descriptors (especially persistence and recurrence time) seem to be sensitive to changes of the $CO_2$ increase. That means a stronger increase of $CO_2$ (e.g. from 1950 on) yields to an decrease of the persistence and increase of the recurrence time. Again it is of utmost importance to note, that we removed the linear trends from each 30 year section of the temperature and EDI data.

2. Thus we can conclude that the natural variability can be approximated by the variability observed before and after the shift. This natural variability is smaller than the shift of the mean.

Concluding, due to the non-availability of preindustrial data we could not really test natural variability vs. natural nonstationarity. But we could show that natural variablity (before and after the shift in 1950) is smaller than the shift, which is probably due to the change in $CO_2$ increase. The mean level shift for the winter extremes of the persistence is about 50% (from 0.2 to 0.1) and for the recurrence time it is about 20% (from 180 to 140 days). Regarding Fig. 8 in the original manuscript we see that changes of the persistence above 50% have been observed (red and cyan regions) and changes of the recurrence time above 20% (red and green). Thus, according to

the sensitivity tests natural variability can most probably be excluded as the sole cause for these changes. Interestingly our significance test also states that these changes are significant with very small p-values. These findings strongly support the results found in our study that changes of the succession of compound extremes are likely to occur in the future due to the increasing $CO_2$ emissions, whereas natural variability plays a minor role.

**One can test numerical variability of the descriptors by constructing appropriate surrogate data. E.g., FT surrogate data generation averages dynamics over whole record randomized, so one can get ranges for random variability of the descriptors in a stationary data.**

We have done this as part of our analysis and will now include the results in the revised version. To construct FT surrogates of our data, we used the MIAAFT algorithm (Venema et al., 2006) which in addition to preserving the original distribution of the data also preserves the auto and cross-correlation of the temperature and precipitation time series. 100 surrogate data sets for the 6 regions used throughout the paper were calculated for the E-Obs data set in the reference period (1971-2000) and their standard deviation taken as the error (by using the exact same regions the values are transferable to later chapter which would not be possible had we chosen a different number of data points). An overview of the errors can be seen in Tab. 1. The errors are fairly similar for all regions and do not differ largely between the two seasons. As in the original manuscript, we will keep on using the ensemble approach for estimating the uncertainty of the descriptors and their climate change signal, but will refer to these MIAAFT estimated errors when discussing the results throughout the paper.

|      | DJF | | | JJA | | |
|------|-------|-------|-------|-------|-------|-------|
|      | P     | R     | E     | P     | R     | E     |
| reg1 | 0.010 | 1.701 | 0.004 | 0.007 | 1.183 | 0.009 |
| reg2 | 0.011 | 2.182 | 0.010 | 0.010 | 2.055 | 0.010 |
| reg3 | 0.010 | 2.563 | 0.005 | 0.009 | 0.923 | 0.007 |
| reg4 | 0.008 | 1.150 | 0.005 | 0.008 | 0.990 | 0.011 |
| reg5 | 0.010 | 2.45  | 0.010 | 0.008 | 1.103 | 0.009 |
| reg6 | 0.007 | 0.797 | 0.004 | 0.009 | 1.150 | 0.009 |

Table 1: Estimation of the error of the descriptors by using MIAAFT surrogates for winter (DJF) and summer(JJA) extremes. Values are calculated for the 6 regions of Fig. 1.

**2  Technical corrections**

**P. 9, last para ....Fig. 4) all regions except Bulgaria... Should not it be France?**

Yes thank you, it should be France

**p 10, 4.2 The statistical treatment should be described in more details: Differences of the ensemble means are plotted, i.e. one get the mean and percentiles for each ensemble, then the difference of means is clearly defined, but what are the percentiles?**

The climate change signal is calculated for each ensemble member separately. What is shown in the plot is the mean difference (gray bar), as well as the median and interquartile range (box) and the minimum/maximum difference (whiskers). We will add a more detailed description in the text so it becomes clearer.

**Is this an appropriate way to evaluate the significance of changes?**

The significance of the changes is determined by the ensemble approach and we think that this is an appropriate way of analyzing significance in this context. Furthermore the changes can be compared to the errors as derived by the MIAAFT algorithm (of the newly added Chapter). Changes are larger then the there derived errors which is an additional indicator of significance. We will mention this additionally in the text. Furthermore, as explained above, the significance test is in accordance to our sensitivity tests. These sensitivity tests have shown that changes of the persistence in the order of 50% (for recurrence time 20%) cannot be achieved by natural variability, but by a shift of the increase in $CO_2$ emissions. Similarly, the significance test states that changes of the persistence in the order of 50% (recurrence 20%) are significant.

**References**

Taylor, R. P., Guzman, R., Martin, T., Hall, G., Micolich, A., Jonas, D., Scannell, B., Fairbanks, M., and Marlow, C.: Authenticating Pollock paintings using fractal geometry, Pattern Recognition Letters, 28, 695–702, 2007.

Venema, V., Meyer, S., García, S. G., Kniffka, A., Simmer, C., Crewell, S., Löhnert, U., Trautmann, T., and Macke, A.: Surrogate cloud fields generated with the iterative amplitude adapted Fourier transform algorithm, Tellus A, 58, 104–120, 2006.

[Figure]

Figure 1: E-Obs descriptors for the reference period (1971-2000). Left side: Descriptors for cold and wet extremes in winter (DJF) (Ta < 10th percentile and Pa > 75th percentile). Right side: Descriptors for hot and dry extremes in Summer (JJA) (Ta > 95th percentile and EDI < 25th percentile). Descriptors were calculated for a moving window over 9 grid points and values assigned to the center grid point. Boxes show the Prudence Regions (http://ensemblesrt3.dmi.dk/quick-look/regions.html).

[Figure]

Figure 2: Descriptors for ECA& D station data for running windows over 30-years (values are assigned to the first year of the 30-year time period.) from 1900-2015. Black lines: cold and wet extremes in winter (DJF) (Ta < 10th percentile and Pa > 75th percentile). Gray lines: hot and dry extremes in Summer (JJA) (Ta > 95th percentile and EDI < 25th percentile).

---

## Author Response (AR1)

**Answer to comments of the referees**

**Compound extremes in a changing climate - a Markov Chain approach**

**K.Sedlmeier, S. Mieruch, G. Schädler and C. Kottmeier**

In the following, we have answered all the comments of the three referees. The comments are highlighted in red color with our answers in black below. In these answers, we give a reference to the line of the latex diff file (attached below) in green where the changes in the manuscript can be seen.

**1 Referee 1**

**Section 3 discussion of results (approx L250 on), it would be good to see some comparison with other research on the persistence of extremes in different regions and possible causes. e.g. Sillmann & Croci-Maspoli 2009, Furrer et al 2010, Photiadou et al 2014. Furrer, E.M., R.W. Katz, M.D. Walter, and R. Furrer, 2010: "Statistical modeling of hot spells and heat waves." Climate Research, 43, 191-205 Photiadou, C., Jones, M., Keellings, D., Dewes, C., 2014. Modeling European hot spells using extreme value analysis. Clim. Res. 58, 193–207. doi:10.3354/cr01191 Sillmann, J., Croci-Maspoli, M., 2009. Present and future atmospheric blocking and its impact on European mean and extreme climate. Geophys. Res. Lett. 36, L10702. doi:10.1029/2009GL038259**

We have added a reference to some of the above mentioned research in the discussion sections (L 571-578). However a direct comparison is difficult as most papers refer to absolute univariate extreme events. Nevertheless follow up studies to analyze the interdependence between atmospheric drivers and the here discussed dynamical aspects of relative extremes would be very interesting as they most likely also have an influence on the latter.

**Similarly a sentence or two comparing the reliability of different models and observations would be good - e.g. CFSR and ERA-40 can be very different. This could be in the data section.**

Thank you for this comment, we included a sentence in the methods section (Sect. 2.1.,L 106 and 2.2 L 132) and at the end of Sect. 4 (L 432) of the revised versionas well as the comment that the detection of differences in observational/reanalysis datasets and models concerning the dynamical behavior of extreme events is an additional interesting application of the method.

**Did you test the significance of the changes in the reference period as well as the future? How did you account for uncertainty in the results?**

Regarding the **uncertainty**, we took advantage of the applied ensemble approach. In Figs. 4 and 5 (of the revised manuscript) we show the results of the ensemble for the reference period, where

we use a box plot for the ensemble: box = ensemble median and interquartile range, whiskers = ensemble minimum/maximum, gray bars: ensemble mean. This information is given in the text caption, to make it clear, we included it in the text under Sect. 4 (L 374ff). Similar box plots accounting for the ensemble uncertainty have been used in Figs. 10 and 11 showing the change signal of the descriptors. Here the changes were calculated for each ensemble member individually- the ensemble mean change is shown by the bar and the interquartile range of the change signal by the whiskers. As can also be seen from the figures we did not account for the uncertainties in the observational E-OBS dataset and consider the observations approximately as the truth. Nevertheless we included an additional section in the revised version where we calculated the error of the descriptors by a FT-resampling algorithm (Sect. 3.3, L 353-372). For this we used the MIAAFT algorithm (Venema et al., 2006) which in addition to preserving the original distribution of the data also preserves the auto and cross-correlation of the temperature and precipitation time series. 100 surrogate data sets for the 6 regions used throughout the paper were calculated for the E-Obs data set in the reference period (1971-2000) and their standard deviation taken as the error (by using the exact same regions the values are transferable to later chapter which would not be possible had we chosen a different number of data points). An overview of the errors can be seen in Tab. 2 (page 12). In comparison to differences between regions and time periods, the error is small but we will include it in the discussions of Sect. 4 and 5. Regarding the **significance** we use the ensemble uncertainty, as mentioned above. In Sect. 5.2 we use the nonparametric Mann-Whitney-Wilcoxon test for the change signal (Figs. 10, 11). The p-values are shown below the bars in the respective figures.

**L338 note about relative extremes - This should really be mentioned in the method section along with how you selected the extremes (e.g. thresholds, and at which level). Possibly a table of extremes would be informative for comparison?**

We introduced a new subsection in the methods section of the revised manuscript where the thresholds and the partitioning of the data are described. This section 2.1. is called The Markov descriptors for two compound extremes. (L 258-273)

**Minor corrections**

We corrected these mistakes, the line numbers of the diff file where the changes were made are given in green.

L3 "the number of occurrences" L 3

L9 types L 9

L11 replace "which are" with "including" L 12

L12 rogue comma before fullstop. L 13

L26 occurrences  L 39

L36 changes in the number of L 39

L46 should this be chaotic attractor?  No

L107 please put into present tense to match the rest of the text. L 116

73  L115 ditto L 117

74  L145 unnecessary comma at start of line. L 160

75  L180 "number of states" L 195

76  L189 and 192 "Thus in the sense of successive compound..." L 207

77  L216 should this be per 100 days?    No, 1000days is correct because this number refers to the

78  total number of days, not only the compound extreme states.

79  L245 maybe say very rare? There are a lot of extremes in that sentence.  L 380

80  L273 highest persistence is L 412

81  Figure 9 caption rogue fullstop before Percentages. now Fig. 7

82

**2   Referee 2**

84  In order to address the first comments of referee 2, we introduced a new section in the revised

85  version called "Sensitivity analysis" (Sect. 3, L 274-371) where we address the spatial and natural

86  variability and analyze the error by means of Fourier-Transform surrogate time series.  Detailed

87  comments can be found below.

88  **one should demonstrate that new descriptors reasonably reflect underlying physical**

89  **mechanisms. Before using any new measure for characterization of ongoing and ex-**

90  **pected climate change, one should investigate their variability in natural conditions.**

91  **The authors use the gridded E-OBS data set, however, they unfortunately chose just**

92  **a few grid points in six different areas.  It is a pity, since the E-OBS data set gives**

93  **an excellent opportunity to study spatial variability of any descriptor which has an**

94  **ambition to characterize the temporal evolution of a physical quantity attributed to**

95  **each grid point. I think the model is reasonably simple to compute full coverage for**

96  **Europe for all three descriptors and map them. The simple visual evaluation would**

97  **indicate if the descriptors reasonably reflects physical reality in the case the maps**

98  **show interpretable smoothly changing patterns. Or, if the maps show just a colored**

99  **grains or a sort of Pollocks paintings, than there is a problem with the descriptor**

100 **and its connections to physical reality.**

101 We thank the referee for that comment and totally agree that new descriptors must be tested for

102 revealing a connection to physical reality. Indeed, we did these tests prior to our analysis, which

103 were also the basis for choosing the regions discussed in this paper. We have calculated a full

104 coverage for the descriptors averaging over 3x3 grid points for the whole area and these maps

105 show interpretable smoothly changing patterns as you can see in Fig. 1. This figure is included

106 and discussed in the revised version of the paper in the newly introduced section (L 279-312)

107 As to Pollock's painting: a map like a Pollock's painting might not be achieved easily for the

108 Markov descriptors. Pollock's paintings are not random and not noise, rather they are in between

109 determinism and noise, they are fractal (Taylor et al., 2007, , and citations therein). Thus, due to

their fractal geometry they have deep underlying mechanisms in common with natural patterns
and hence also with our atmospheric time series.

**While E-OBS data set can be used to test spatial variability, ECA&D station data set offers a number of long-term records in which temporal variability can be tested. So one can relate the change of the introduced descriptor due to climate change to their changes due to natural variability in preindustrial era. Real long-term records would reflect natural variability due to natural nonstationarity.**

This is a good suggestion. Unfortunately, there is only one station with a **continuous** (without missing values) temperature and precipitation record (starting in 1887) available from the ECA&D data set. Further, only a few stations within Germany have available **continuous** time series starting in 1900. Nevertheless we calculated the descriptors for a combined time series of the available 7 stations in Germany for running windows of 30 years starting in 1900. The combination of the time series is necessary in order to fulfill the stationarity criteria explained in Section 2.3. (non zero entries of the transition probability matrix and stationarity of the time series). It is important to note that we removed all linear trends for each 30 year section seperately as it has been done in the rest of the paper. The resulting time series of the descriptors are shown in Fig. 2 for both winter (black) and summer (gray) extremes. These results are included in Sect. 3, L 313-352) in the revised version of the paper. The stations used are listed in the data section (Tab. 1). Especially for the persistence and recurrence time, a clear shift is visible between 1930 and 1950. This time range is not preindustrial, but the crucial point is that the observed shift coincides with a globally oberved shift in the increase in CO2 around 1950 (http://www.ldeo.columbia.edu/~spk/Research/AnthropogenicCarbon/images/ddic_uptake_hist.png). Thus from this finding we observe two main points:

1. The descriptors (especially persistence and recurrence time) seem to be sensitive to changes of the $CO_2$ increase. That means a stronger increase of $CO_2$ (e.g. from 1950 on) yields a decrease of the persistence and increase of the recurrence time. Again it is of utmost importance to note, that we removed the linear trends from each 30 year section of the temperature and EDI data.

2. Thus we can conclude that the natural variability can be approximated by the variability observed before and after the shift. This natural variability is smaller than the shift of the mean.

Concluding, due to the non-availability of preindustrial data we could not really test natural variability vs. natural nonstationarity. But we could show that natural variablity (before and after the shift in 1950) is smaller than the shift, which is probably due to the change in $CO_2$ increase. The mean level shift for the winter extremes of the persistence is about 50% (from 0.2 to 0.1) and for the recurrence time it is about 20% (from 180 to 140 days). Regarding Fig. 10 we see that changes of the persistence above 50% have been observed (red and cyan regions) and changes of the recurrence time above 20% (red and green). Thus, according to the sensitivity tests natural variability

can most probably be excluded as the sole cause for these changes. Interestingly our significance test also states that these changes are significant with very small p-values. These findings strongly support the results found in our study that changes of the succession of compound extremes are likely to occur in the future due to the increasing $CO_2$ emissions, whereas natural variability plays a minor role.

**One can test numerical variability of the descriptors by constructing appropriate surrogate data. E.g., FT surrogate data generation averages dynamics over whole record randomized, so one can get ranges for random variability of the descriptors in a stationary data.**

We have done this as part of our analysis and will now include the results in the revised version in the newly introduced section (Sect. 3.3,L 353-371). To construct FT surrogates of our data, we used the MIAAFT algorithm (Venema et al., 2006) which in addition to preserving the original distribution of the data also preserves the auto and cross-correlation of the temperature and precipitation time series. 100 surrogate data sets for the 6 regions used throughout the paper were calculated for the E-Obs data set in the reference period (1971-2000) and their standard deviation taken as the error (by using the exact same regions the values are transferable to later chapter which would not be possible had we chosen a different number of data points). An overview of the errors can be seen in Tab. 2. The errors are fairly similar for all regions and do not differ largely between the two seasons. As in the original manuscript, we will keep on using the ensemble approach for estimating the uncertainty of the descriptors and their climate change signal, but refer to these MIAAFT estimated errors when discussing the results throughout the paper.

**P. 9, last para ....Fig. 4) all regions except Bulgaria... Should not it be France?**

Yes thank you, it should be France we have changed that. (L 440)

**p 10, 4.2 The statistical treatment should be described in more details: Differences of the ensemble means are plotted, i.e. one get the mean and percentiles for each ensemble, then the difference of means is clearly defined, but what are the percentiles?**

The climate change signal is calculated for each ensemble member separately. What is shown in the plot is the mean difference (bar), as well as the interquartile range (whiskers). We added a sentence at the beginning of Sect. 5.2 (L 469ff) of the revised version so it becomes clearer and have clarified it in the captions of Figs. 10 and 11.

**Is this an appropriate way to evaluate the significance of changes?**

The significance of the changes is determined by the ensemble approach and we think that this is an appropriate way of analyzing significance in this context. Furthermore the changes can be compared to the errors as derived by the MIAAFT algorithm (of the newly added Chapter). Most changes are larger then the there derived errors which is an additional indicator of significance.

We additionally mentioned this in the text of the revised manuscript. Furthermore, as explained above, the significance test is in accordance to our sensitivity tests. These sensitivity tests have shown that changes of the persistence in the order of 50% (for recurrence time 20%) cannot be achieved by natural variability, but by a shift of the increase in $CO_2$ emissions. Similarly, the significance test states that changes of the persistence in the order of 50% (recurrence 20%) are significant.

The explanation has been expanded in the revised version, see L 482ff )

**3   Referee 3**

**The authors should refer other approaches like the geostatistical analysis of spatially distributed extremes (Neves 2015). That is important because extremes have themselves some spatial organization.**

We included this in the introduction of the revised version. (L 36ff)

**There is no clear justification for the choice of the 6 box-regions and their size (6x6 grid points). Why they are representative of the PRUDENCE regions? Some minimal study about the spatial robustness of the Markov diagnostics should be presented. For example, does the results keep similar or change substantially when contiguous boxes are considered? The ideal should be to present maps of the diagnostics throughout Europe.**

We thank the referee for that comment, because indeed, we have performed tests on the robustness of the Markov descriptors, which are the basis for the decision to use the actual 6 box-regions. The crucial point, why we have used the 6 box-regions is to achieve that each region contains the exact same amount of grid points / data points. This is of utmost importance for the comparison of the regions, otherwise, if the regions have been chosen with differing sizes no consistent comparison would be possible due to the fact that the Markov descriptors depend on the underlying sample size of the used data. To account for the spatial robustness we calculated the Markov descriptors for every grid point in Europe and visualised the results on a maps. From these maps we have seen that the Markov descriptors vary in general not strongly within the prudence regions. Accordingly we have chosen the 6 box-regions within the Prudence regions, which are representative for the respective region, based on the results of the grid point maps. In the revised we included the maps showing the grid point results (see Fig. 1 in the revised version.)

**In the entropy definition H (eq. 7), log(1/m) must be replaced by log(m) so that H equals 1 for a random system without memory (all probabilities pij=1/m).**

Thank you for the comment but our definition corresponds to those of other papers (see eg. Hill et al., 2004). Maybe you have missed the - sign at the beginning or the "‘/"’ sign in the equation

(log(1/m) is the same as -log(m))? By using log(m) we would get negative entropies with our formula.

**Line 189: Authors claim that H between 0 and 1 is an identification of deterministic chaotic behavior. However that condition is necessary but not a sufficient condition for chaos. Authors shall carefully rephrase the paper by taking that into account.**

We agree with this comment and have rephrased the follwing sentence:

*The dynamics of complex chaotic systems lie in between these limits, thus the entropy can be used to identify and characterize complex dynamics like deterministic chaos, which is not possible with standard linear methods*

by

*The dynamics of complex chaotic systems lie in between these limits, thus the entropy can give a hint to underlying complex dynamics like deterministic chaos, which is not possible with standard linear methods. To really test for deterministic choas other methods, based on state space reconstruction (e.g. estimating the correlation dimension, Lyapunov exponents etc.) to find strange attractors, are more suitable.* (L 202ff)

and references to the chaotic behavior accordingly throughout the revised version of the paper.

**Line 197: Authors say The reason for this is that the CO2 forcing is the only difference. . .. In fact, decadal variability is also likely. That sentence must be weakened by replacing the only by the main difference beyond the natural decadal variability.**

No, because the crucial point is that this sentence (L 197) refers to the **model runs** (cf. line 198). The decadal variability of the model is not intrinsically changing with time. The only difference between the model runs in the past and in the future is the $CO_2$ forcing. Thus, changes of the decadal variability are of course possible, but the only reason is a changing $CO_2$ forcing. See L 15ff and the results in Sect. 3.2

**Eq. 8 explain the meaning of the bar and subscripts rm.**

Yes, we will do so and have also included a more detailed explanation of the EDI in the revised version (also see next comment). The bar in equation 8 stands for the climatological mean - $\overline{EP_{d,rm}}$ refers to the climatological mean state of EP corresponding to day d, where the climatological mean is calculated by a running mean of rm days over the 30 years of the respective time period. (Sect.2.4., L 245ff)

247 **Line 234: Droughts may have different time scales from months to years. That is**
248 **the reason for defining the SPI (Standard Precipitation index) (McKee et al. 1993).**
249 **The presented EDI is appropriate for annual scaled droughts. Add this comment to**
250 **the text. Moreover the EDI has its own annual cycle since the precipitation weights**
251 **contributing to EDI are larger near the Julian day d. Does the annual cycle of EDI**
252 **was removed?**

The EDI does not have an annual cycle as this is intrinsically removed by the method
(e.g. http://atmos.pknu.ac.kr/~intra2/eng.calculation.htm). In the equation:

$$EDI_d = \frac{EP_d - \overline{EP_{d,rm}}}{\sigma \left( EP - \overline{EP} \right)_d} \tag{1}$$

253 $\left( \overline{EP} \right)_d$ refers to the climatological mean state of EP and is calculated for each day as the 5day
254 running mean over the 30years of the respective time period. Thus, by subtracting $\left( \overline{EP} \right)_d$ from
255 EP, the annual cycle is removed. We are sorry that this did not become clear and have include a
256 more thourough explanation of the EDI in the methods section of the revised version (Sect.2.4,
257 L 251ff) and clearly state that the annual cycle is removed by the method.

258 Furthermore, the EDI is not only appropriate for annual scaled droughts. Since it is calculated
259 from daily values, it is also able to detect droughts of shorter lengths. It is highly correlated to the
260 soil moisture. For example if there is heavy rain on August 1st and September 30th, the EDI can
261 detect a water deficit in between these two dates, whereas a monthly indice would not detect this
262 (see Byun and Wilhite, 1999).

263 **L235-238 Does temperature anomalies (Ta) and precipitation anomalies (Pa) refer to**
264 **daily Ta and daily Pa with respect to the respective annual cycle. Please clarify. Add**
265 **a sentence about the number of categories of the Markov chain and what categories**
266 **of the compound attractor were considered? I suppose that authors have considered**
267 **2 parameters with a partition of 2 categories each. Confirm that at this stage for the**
268 **sake of the paper understanding.**

269 Yes, the Ta and Pa refer to daily temperature and precipitation anomalies with respect to the annual
270 cycle. And we have considered 2 parameteres with a partition of 2 categories each which we then
271 combined to a 4 state symbolic sequence. We added a new subsection to the methods section (

272 **Fig. 3 In the recurrence plot I cannot see the black triangle for region 1.**

273 thank you for the notice, we have changed that. (Fig. 5)

**Fig. 4 In the caption, descriptors changes refer to changes in the period 1981- 2010 with respect to 1951-1980? Rewrite it in a clearer way.**

We have replaced *1951-1980 vs 1981-2010.* by *changes between the time periods 1951-1980 and 1981-2010* (see Figs. 6, 7,10,11)

**4 Additional changes**

We have recalculated the entropy as we detected an error in the original manuscript. Therefore the entropy values are changed in the revised version. Furthermore we have changed the results for the summer extremes from TA > 95th percentile to TA>90th percentile. This way the summer and winter extremes yield the same number of univariate compound extreme events and the behavior of the two compound events can be better compared.

[revised manuscript text omitted]

---

## Author Response (AR2)

**Answer to comments of the referees**

**Compound extremes in a changing climate - a Markov Chain approach**

K.Sedlmeier, S. Mieruch, G. Schädler and C. Kottmeier

In the following, we have answered the comment of the referee (comment in in red, answer below). The marked-up manuscript is attached below.

**The authors adequately responded all questions, including the spatial variability of the proposed descriptors. Here, however, I do not fully understand the procedure of 9-gridpoint moving windows: how it was technicaly done, which neighbouring gridpoint are included and how the window is moved and, most importantly, if this procedure could induce spatial smoothness which would not be present wihtout this procedure. I propose an explanation of this procedure and its consequences as an optional revision which the authors could do in order to improve the understandability of their data processing procedure.**

The 9 point moving window was introduced in order to fulfill the criteria of the Markov method (stationarity and non-zero entries of the transition probability matrix, see Sec. 2.3) which is not given for the whole area when calculating the descriptors from single gridpoints. Using single gridpoints also shows smooth results and the general spatial pattern of the descriptors is not altered by using this moving window. Therefore we used 9 gridpoints, thus enlarging the time series to avoid not meeting the criteria mentioned above. We calculated the descirptors for each grid point using the time series of that grid point and the 8 boardering gridpoints. The time series of the resulting 9 grid points are partitioned seperately and then merged together before calculating the descriptors. We added this explanation in the revised manuscript (lines 275ff).

[revised manuscript text omitted]